# Satellite retrievals of dust aerosol over the Red Sea and the Persian Gulf, 2005-2015

Jamie R. Banks[1,2], Helen E. Brindley[3], Georgiy Stenchikov[4], and Kerstin Schepanski[1]

[1]Leibniz Institute for Tropospheric Research, Leipzig, Germany.
[2]Space and Atmospheric Physics Group, Imperial College London, London, UK.
[3]Space and Atmospheric Physics Group, and NERC National Centre for Earth Observation, Imperial College London, London, UK.
[4]Division of Physical Sciences and Engineering, King Abdullah University of Science and Technology, Thuwal, Saudi Arabia.

*Correspondence to:* Jamie R. Banks (banks@tropos.de)

**Abstract.** The inter-annual variability of the dust aerosol presence over the Red Sea and the Persian Gulf is analysed over the period 2005-2015. Particular attention is paid to the variation in loading across the Red Sea, which has previously been shown to have a strong, seasonally dependent latitudinal gradient. Over the eleven years considered the July mean 630 nm aerosol optical depth (AOD) derived from the Spinning Enhanced Visible and InfraRed Imager (SEVIRI) varies between 0.48 and 1.45 in the southern half of the Red Sea. In the north the equivalent variation is between 0.22 and 0.66. The temporal and spatial pattern of variability captured by SEVIRI is also seen in AOD retrievals from the MODerate Imaging Spectroradiometer (MODIS) but there is a systematic offset between the two records. Comparisons of both sets of retrievals with ship and land-based AERONET measurements show a high degree of correlation with biases of $< 0.08$. However, these comparisons typically only sample relatively low aerosol loadings. When both records are stratified by AOD retrievals from the Multi-angle Imaging SpectroRadiometer (MISR) opposing behaviour is revealed at high MISR AODs ($> 1$), with offsets of +0.19 for MODIS and -0.06 for SEVIRI. Similar behaviour is also seen over the Persian Gulf. Analysis of the scattering angles at which SEVIRI and MODIS typically make retrievals in these regions suggests that assumptions concerning particle sphericity may be responsible for the differences seen.

## 1 Introduction

Desert dust aerosols have a substantial influence on the atmospheric environment of the Middle East (e.g., Osipov et al., 2015; Kalenderski and Stenchikov, 2016), and of the maritime environment of the Red Sea (e.g., Jish Prakash et al., 2015). The latter is pinned between two of the largest sandy desert regions in the world, the Sahara of northern Africa to the west, and the Arabian and Syrian Deserts to the east. Of broader scientific, environmental and cultural interest in this region is the biodiversity of the Red Sea and its substantial coral reef systems, which are at risk of increasing global ocean temperatures and which have experienced substantial degradation and 'bleaching' in recent times (e.g., Cantin et al., 2010; Pandolfi et al., 2003). Over land, the Middle East and the Arabian Peninsula have seen a recent increase in dust activity over the past ten years (Notaro et al., 2015), a situation aggravated by drought conditions in the semi-arid regions of Syria and Iraq in the north (the Levant). It is an

open question as to whether this increase in activity has propagated through to increased atmospheric dust loadings over the Red Sea, and similarly over the Persian Gulf to the east of the Arabian Peninsula.

Despite the region's vulnerability to the impacts of desert dust aerosol, it is relatively little monitored compared to the Mediterranean to the north and to the neighbouring desert regions. The Sahara, for example, has been investigated over the past ten years by numerous field campaigns such as AMMA, SAMUM, GERBILS, and Fennec (Redelsperger et al., 2006; Heintzenberg, 2009; Haywood et al., 2011; Ryder et al., 2015). Aerosol Robotic Network (AERONET) sun-photometers (Holben et al., 1998), which are part of a global network aiming to provide long-term observations of atmospheric aerosol loading, are currently only located on the Red Sea coast at Eilat in Israel (at the far north of the Red Sea) and the King Abdullah University of Science and Technology (KAUST) in Saudi Arabia at 22.3°N, and the latter was only established in February 2012.

Nevertheless, Brindley et al. (2015) (subsequently denoted B15) recently investigated the performance of satellite retrievals over the Red Sea, making use of ship cruise data taken under the framework of the Maritime Aeronet Network (Smirnov et al., 2009). They found close agreement between dust aerosol retrievals from the Spinning Enhanced Visible and Infrared Imager (SEVIRI) and the MODerate resolution Imaging Spectroradiometer (MODIS) instruments. SEVIRI-MODIS AOD offsets were found to be +0.02, with correlations of 0.93 and 0.95 for the two sets of cruises considered, with similar statistics obtained when considering the agreement between the two sets of retrievals and the ship-based measurements. The same study identified a strong seasonality in dust loading over the Sea and a particularly marked latitudinal gradient in summertime dust AOD. This differential loading was shown to propagate to significantly enhanced surface radiative cooling in the southern part of the basin relative to the north.

However, the B15 study was limited to a five year period from 2008-2012 and used MODIS 'Collection 5' data, which have since been superseded by 'Collection 6' (Levy et al., 2013). More critically, the ship-based cruise data used to evaluate the retrievals did not sample conditions typical of the summertime southern Red Sea. Banks et al. (2013) identified systematic differences between dust retrievals over the Sahara, influenced by such factors as dust loading, atmospheric moisture, and surface emissivity. It is to be expected that over a longer time period and given sufficiently high dust loadings the retrievals over the Red Sea may also exhibit systematic differences.

Hence in this paper we extend the B15 analysis and investigate in greater detail the latitudinal structure of dust aerosol presence over the Red Sea, as inferred from SEVIRI and MODIS Collection 6 retrievals. The 11-year pattern in dust activity is presented and discussed, with reference to both the temporal and spatial structure. We make use of all the ship-based cruises now available over the Red Sea (from 2010-2015) and the KAUST AERONET measurements to evaluate the SEVIRI and MODIS AOD retrievals. To deepen our understanding of the differences that we find between the SEVIRI and MODIS retrievals, we also make use of AOD measurements from the Multi-angle Imaging SpectroRadiometer (MISR) satellite instrument, and explore the performance of the retrievals over the Persian Gulf in comparison with that over the Red Sea.

## 2 Satellite retrievals

Examining retrievals over sea/ocean only, the 'wider' region investigated here is within the bounds of 12-30°N, 32-56°E, encapsulating both the Red Sea and nearby waters including the Persian Gulf and parts of the Arabian Sea. This wider region is used to set activity over the Red Sea into context. Meanwhile the Red Sea itself ('the Sea') is located within the region
bounded by 12-30°N, 32-44°E. Egypt, Sudan, Eritrea and Djibouti border it to the west, Israel and Jordan to the north, and Saudi Arabia and Yemen to the east. The southernmost point at 12.58°N is the 'Bab-El-Mandeb', a narrow strait between Yemen and Djibouti. At the northern end the Red Sea again narrows considerably where it splits into two, the Gulfs of Suez (west) and Aqaba (east), ending at 29.97°N. The total length of the Red Sea from the Bab-El-Mandeb to the northern end of the Gulf of Suez is approximately 2200 km, but the maximum width of the Sea is less than 400 km. This narrow width of the Sea
is such that it can effectively be subdivided into latitude bands to describe the various sub-domains within the basin. A map of the region and its topography is presented in Figure 1. The Persian Gulf is a similarly nearly-enclosed sea which is susceptible to dust activity, and is located to the east of the Arabian Peninsula: in the context of this study it is bounded by the longitudes 48 and 56°E, and the latitudes 24 and 30°N.

### 2.1 SEVIRI AOD retrievals

In geostationary orbit above the equatorial east Atlantic the Meteosat Second Generation (Schmetz et al., 2002) series of satellites carry the SEVIRI instruments. Every 15 minutes SEVIRI images the facing disc of the Earth in eleven visible and IR channels, viewing Africa, Europe, much of the Middle East, and parts of the Americas. A twelfth channel provides higher resolution visible image data over a more limited geographical area. At nadir the spatial sampling rate is 3 km, towards the limb of the Earth's disc over Saudi Arabia and the Red Sea a more typical spatial sampling rate is ∼4.5 km. Over the Red Sea
the SEVIRI viewing zenith angle is ∼50°, whereas over the Persian Gulf the viewing zenith angles are even higher, exceeding 62°.

SEVIRI retrievals over ocean (Brindley and Ignatov, 2006) make use of the visible reflectance channels at 0.6 and 0.8 $\mu$m, and the near-IR reflectance channel at 1.6 $\mu$m. A look-up table (LUT) approach is used to simulate the oceanic surface reflectance ($\rho_{sfc}$), which contains contributions from ocean glint, whitecaps, and underlight. The LUT is also used to simulate
the aerosol contribution to the reflectance, as a function of aerosol optical depth (AOD). Output AOD retrievals are provided at each of 630, 830, and 1610 nm, in this study we make use of the 630 nm AOD retrievals.

Aerosol retrievals are only taken during the daytime when there is sufficient solar illumination, above a solar elevation angle of 20° and below a viewing zenith angle of 70°. Sunglint off the oceanic surface is also an issue for the visible retrievals, at those angles where the SEVIRI detector is affected by specular reflection off the sea surface, since the enhanced apparent brightness
of the sea surface reduces the contrast with any aerosol present. The minimum allowable glint angle is 30°. Retrievals are also only carried out in the absence of cloud: cloud must first be flagged (Ipe et al., 2004), although this procedure is known to be conservative and may flag dust as cloud (De Paepe et al., 2008). This can be corrected using IR channels to carry out dust flagging (Brindley and Russell, 2006), in order to reinstate dusty pixels previously misidentified as cloud.

Previous validation efforts of SEVIRI retrievals over the Red Sea against ship-based measurements (B15) in 2011 and 2013 indicate SEVIRI biases for the two individual years of +0.03 and +0.04, and root-mean-square differences (RMSD) for both years of 0.06. The maximum AOD measured at 675 nm during this time period was less than 1.1.

## 2.2   MODIS retrievals of AOD

Carried by the polar-orbiting NASA Aqua and Terra satellites, the MODIS instruments observe any point on the Earth's surface nearly twice-daily, at night and during the day. In daylight the local equator crossing times are ∼1030 LT for Terra and ∼1330 LT for Aqua. Three algorithms have been developed to retrieve aerosol quantities and properties: two of these are often referred to as the 'Dark Target' (Tanré et al., 1997; Levy et al., 2003; Remer et al., 2005; Levy et al., 2007) algorithms, which are typically used to retrieve aerosol over water and over dark and vegetated surfaces. The two algorithms are distinct for

land and for ocean, and make different assumptions about surface and aerosol properties. Over ocean, the algorithm uses six visible and near-IR MODIS channels with wavelengths between 550 and 2110 nm, and makes use of lookup-tables which relate measured reflectances with angles, aerosol size distributions and optical depths; this is the MODIS AOD product considered in this study. Over brighter, e.g. desert surfaces, the 'Deep Blue' (Hsu et al., 2004) algorithm has enhanced performance for retrieving aerosol, using the near-UV channels of the instrument. At nadir the MODIS spatial resolution ranges from 250 m

to 1 km depending on channel, becoming larger towards the edges of the MODIS swath. AODs are reported, among other wavelengths, at 550 nm, and at a spatial resolution of ∼10 km in the L2 dataset. Other relevant aerosol information retrieved by MODIS includes the Ångström coefficient, useful for re-scaling the MODIS AOD retrievals to the same wavelength as the SEVIRI AOD retrievals.

In this study we use the 'Collection 6' (C6) algorithm, the most recent iteration of the MODIS AOD dataset (Levy et al.,

2013; Hsu et al., 2013; Sayer et al., 2013), using Dark Target retrievals over ocean, from both the Terra and Aqua satellites. As opposed to the previous Collection 5 Dark Target retrievals, the philosophy behind C6 has been described as a 'maintenance and modest improvement' upgrade by Levy et al. (2013), involving updates to such processes as calibration (which affects the measured channel reflectances), cloud-masking, geolocation and land/sea discrimination, and assumptions in radiative transfer modelling. Additionally over ocean there are changed assumptions about surface wind speed (which affects ocean

glint patterns) and 'sedimented' regions, e.g. treatment of river-outflow or shallow coastal sea regions. Given the long length of coastline with respect to the area of the Red Sea, this latter update may have implications for the retrieval of dust in this region. Pixels retrieved may have four levels of quality-assurance (QA), from 0 (no confidence) - 3 (high confidence). In this paper we use pixels with QA values of 1-3, as recommended by Remer et al. (2008) for MODIS AOD retrievals over ocean. As for retrieval performance, Levy et al. (2013) claim that 76% of MODIS C6 ocean points fall within an 'Expected Error' envelope

of (+(0.04+10%),-(0.02+10%)) with respect to AERONET AOD measurements, asymmetric due to high MODIS biases at low AOD. By contrast 71% of MODIS C5 ocean points fall within this envelope.

## 2.3 MISR retrievals of AOD

The MISR instrument is also carried onboard Terra, and AOD retrieval products are also available from this instrument (Kahn et al., 2009; Abdou et al., 2005). With a swath width of ∼400 km the MISR track is much narrower than that of MODIS (2330 km), however MISR has the advantage of multi-angle scanning, with nine cameras viewing the Earth in the along-track direction between ±70.5° and including one at nadir. Retrieved AODs and estimates of aerosol properties in the MISR version 22 dataset are reported at 17.6×17.6 km spatial resolution, at wavelengths of 446, 558, 672, and 866 nm. The LUT in this algorithm includes eight component particle types subdivided into 74 aerosol mixtures (Kahn et al., 2010; Kalashnikova et al., 2005). Of the eight components, two are non-spherical dust analogues, with one being medium-mode aggregated angular shapes, while the other represents coarse mode ellipsoids. This treatment of non-spherical particles is a distinct advantage that the MISR aerosol retrieval has over both SEVIRI and MODIS Dark Target, which both treat aerosols purely as spheres, an assumption which is questionable for mineral dust. The MISR retrieval is not unique in accounting for non-spherical particles, the Deep Blue algorithm also does not assume particle sphericity. Similarly as with MODIS, the MISR team quote that 70-75% of the AOD retrievals fall within an envelope defined by the larger of 0.05 or 20% of AERONET AOD (Kahn et al., 2010). Within this range, Kahn et al. (2010) report that at higher AODs the MISR retrieval often underestimates the AOD especially over land, due in part to overestimated single-scattering-albedo in the retrieval algorithm.

## 3 A decade of dust activity over the Red Sea

An advantage of the duration that MSG-SEVIRI and the MODIS instruments have now been in orbit for is that we can set the inter-annual variability of dust activity into context, and extend the five-year record presented by B15. It has been noted before that the southern part of the Sea is often dustier than the north, for example by B15, and we extend this previous work by investigating the latitudinal variations in dust loading over the Sea over a longer time period. Figure 2 shows a timeseries of monthly mean SEVIRI AODs at 630 nm over the Red Sea from 2005-2015. The Sea is here subdivided into 'north' and 'south', divided by the line of 20°N which splits the Sea into very closely equal regions by area (13,725 pixels in the north, 13,915 in the south).

The annual cycle is clear and consistent: dust activity peaks in summer, particularly in July, and is concentrated in the southern half of the Sea. The north has an annual cycle, generally peaking in summer, but the amplitude is much less pronounced than it is in the south. Looking for exceptions to the general pattern, 2015 is a somewhat unusual year in that there is no one dominant peak, instead dust activity is spread between the months of April, June, and August. In general the annual cycle is as described by Figure 5 of B15, but we see further interesting features in the breakdown into halves of the Sea and in the extension of the record. Looking at the southern end of the Sea, 2005-2007 is a relatively quiet period, transitioning during 2008 to the dusty period of 2009-2013. This may be associated with the consequences of the Levantine drought period which resulted in enhanced dust activity over the Arabian Peninsula from 2008/2009 onwards (Notaro et al., 2015). Latterly however, the enhanced summer dust activity seems to have tailed off in 2014 and 2015.

Only in July 2010 and August 2015 does the mean AOD in the north exceed 0.5. In the south, dust flows can be trapped above the Sea due to the mountainous topography to either side in both Africa and Arabia (see Figure 1). By contrast in the north the relatively flat topography of the land to either side is less of a hindrance to dust transport and hence dust has less of a tendency to persist for extended periods of time in the atmosphere above the northern part of the Sea.

Spatially on the decadal timescale, as indicated by Figure 3, a sharp gradient in dust loading across the Sea is most strongly apparent in summer (represented here by July). This is also the only season when there is any significant dust activity over the Arabian Sea. For the most part, this timescale disguises specific geographical features. An exception to this is that dust outflows into the southern Red Sea from the Tokar Gap in Sudan (e.g., Prospero et al., 2002; Jiang et al., 2009; Kalenderski and Stenchikov, 2016) are readily apparent in summer (Figure 3(c)), just south of the 20° line of latitude. As

well as being a dust source, the mountain-gap wind jet in this location during summer affects the eddy mixing and circulation of the Sea itself (Farley Nicholls et al., 2015). Further south there are more dust sources on the Eritrean coastal plain, which contribute to the high southern dust loading, but in general much of the dust actually originates from central Arabia and Sudan.

Cloud cover only tends to have a significant effect on the retrieval quantity in summer for one or two months around July during the monsoon period (e.g., Patzert, 1974), when there is persistent cloud presence in the south, coincident with the typical

maximum of dust activity: the fraction of successful retrievals with respect to the total number of attempted retrievals over the southern half of the Sea can be as low as 20%. Cloud cover in the north is typically quite sparse year-round.

### 3.1   The latitudinal gradient in dust loading

B15 postulate that the summertime north-south gradient in dust loading over the Red Sea may have important consequences for atmospheric and oceanic circulation patterns. Exploring this gradient in more detail, we find that the imbalance between

the AODs in the northern and southern parts of the Red Sea basin has a marked inter-annual variability. In order to analyse this further, we subdivide the Sea into nine 2° latitude bands from 12 to 30°N. Monthly averages are then made on the cloud-free SEVIRI pixels of the Sea within these latitude ranges. The grid cells must be sea-only, so the narrow stretches of the Gulfs of Suez and Aqaba (28-30°N), and the narrow, island-studded section of the southernmost Red Sea (12-14°N) have very few retrievals.

Figure 4 describes the latitudinal gradient in SEVIRI AOD over the Sea for the four months of January, April, July, and October, for each year of the eleven-year period. In winter and autumn the gradient is very flat, with a low mean AOD of ∼0.3 across the Sea, and very little inter-annual variability. July, by contrast, has a significant disparity between north and south, with southern AODs marked by a high level of inter-annual variability: for example, at 15°N this varies from ∼0.5 in 2007 to ∼1.7 in 2009. Such variability is not seen in the north, with July 2010 being the sole outlier with relatively high AODs up to

∼0.7 at 23°N. Over the eleven-year period April tends to hold the same 'flat' pattern as January and October, although unlike these months there are two Aprils, 2015 and to a lesser extent 2008, which break away from the pattern: April 2015 especially is marked by substantial dust loading in the south with band mean AODs of up to ∼0.7 between 14 and 20°N, due to a handful of large Arabian dust storms which transported large quantities of dust into the atmosphere of the southern Red Sea.

Where does dust over the Red Sea originate? To derive a quantitative estimate of dust transport over the Red Sea, fields of dust concentration and winds were taken from the Monitoring Atmospheric Composition and Climate (MACC) reanalysis data set (Inness et al., 2013) in order to calculate monthly mean (January, April, July, and October) zonal dust transport fluxes for 2007 and 2009, the 'low' and 'high' summertime AOD years revealed by Figure 4. The MACC reanalysis data set includes a 4D-Var assimilation scheme, and the satellite retrieval data used for aerosols are the MODIS Dark Target AODs (Cuevas et al., 2015). MACC provides fields of atmospheric parameters and aerosol concentrations at 6-hourly resolution. The simulation data used were obtained for each model level and at 1° horizontal resolution. Zonal dust fluxes (M) were then calculated following Equation 1 for each level, $k$, and at each grid cell in the middle of the Red Sea, $i$:

$$M_{i,k} = c_{i,k} * u_{i,k}. \tag{1}$$

Here $c$ is the total dust concentration (particle size: 0.03 - 20 $\mu$m) and $u$ is the zonal wind. As dust originating from both North African and Arabian dust sources is contributing to the dust burden over the Red Sea basin, zonal dust fluxes are calculated separately for eastward and westward transport directions. The computed dust mass fluxes for 2007 and 2009 are presented in Figure 5 by latitude, analogously to Figure 4, and presents fluxes in the eastward and westward directions, subdivided also by dust layer height range. Panels (a) and (b) represent the transport between 2-15 km in altitude, (c) and (d) are between 0-2 km. A caveat to include here is that for simplicity we only compute the zonal dust transport, and hence it is important to note that there will also be a meridional component to the dust transport that is not considered within this analysis. The more significant dust transport is, however, in the zonal direction between the two deserts to the west and east.

Large-scale dust events from Arabia and Sudan are major contributors to the dust loading over the southern Sea in July, along with local sources from Eritrea and the Tokar Gap. Not all Arabian dust events reach the southern Red Sea, a major barrier to dust transport in the south-western Arabian Peninsula is a chain of mountains along the coast, exceeding 2000 m in height. Hence dust which crosses this region must either be lofted above such altitudes or make its way funnelled through valleys and mountain passes, and in consequence Arabian dust is only present over much of the Red Sea at altitudes > 2 km as in Figure 5(b). By contrast, dust outflow from the Tokar gap in Sudan (18-19°) provide up to 0.6 g m$^{-1}$ s$^{-1}$ from Africa predominantly in the lowest layer of the atmosphere (Figure 5(c)): this is consistent spatially with the AODs in Figure 3(c), as well as with dust simulations carried out by Kalenderski and Stenchikov (2016) who also noted the low altitude of Sudanese dust in this region. Figure 5(c) also shows the signature of Eritrean coastal dust sources in the far south, principally active in summer.

Outside of summer, there is less simulated dust transport, and what there is has a tendency to be transported at higher altitudes. April is the outlier to this, when in both years there is a very strong simulated flow of dust travelling eastwards from Africa over the north of the Sea. This is a well-known feature in spring when Sharav cyclones along the African Mediterranean coast emit dust from the north-eastern Sahara and transport these dust outbreaks to Saudi Arabia and the Middle East (e.g., Abed et al., 2009; Notaro et al., 2013). These track from west to east at speeds often greater than 10 m s$^{-1}$ (e.g., Alpert and Ziv, 1989), hence due to quick transit times across the Sea this does not result in particularly elevated dust AODs in the north (Figure 4(b)).

## 4 Validation over the Red Sea using surface-based sun-photometer measurements

In order to verify these patterns in the SEVIRI dust observational record over the Red Sea it is vital to validate and compare these retrievals against other data sources, such as recent ship cruise data. From October 2010 to September 2015 multiple research ship cruises organised by KAUST have taken place on the Red Sea, principally based out of KAUST itself, and

5 Georgiy Stenchikov is the PI of the ship-based aerosol observation campaign. Aerosol measurements were made using a handheld Microtops sun-photometer, carried out following the framework of the Maritime AERONET program of ship-based aerosol measurement campaigns (Smirnov et al., 2009). B15 carried out validation of SEVIRI and MODIS retrievals using these data, but only for 2011 and 2013, and only for MODIS Collection 5 data. We repeat this process here using the updated Collection 6 data, and including 2010, 2012, 2014, and 2015. As of the end of 2015 there are 23 ship-cruise legs available in

the Red Sea, at Level 2 quality. Otherwise the same co-location criteria were used: all three instruments must make a successful retrieval, averaged for satellite pixels within a 50 km radius of the ship and for retrievals within an hour either side of the sun-photometer measurement. As stated earlier, we only use MODIS AOD retrievals with QA values from 1-3. For consistency with the SEVIRI retrievals the AERONET and MODIS AOD values are scaled to a wavelength of 630 nm, using the retrieved Ångström exponent. B15 found 111 co-locations, over the expanded dataset for the Red Sea we now find 255 co-locations.

Note an imbalance in the number of measurements between the northern ($> 20°$N) and southern parts of the Sea: there are 220 co-locations in the north, compared to just 35 in the south. Due to political considerations it was more difficult for the ship cruises to spend much time at the southern end of the Sea. Validation statistics are listed in Table 1.

During this period, few major dust events have been observed from the ship-based measurements over the Sea. The co-located maximum measured AOD at 630 nm by the sun-photometers at sea level was 0.86 in July 2011, at a latitude of 26.3°N.

The mean AOD measured was 0.28 with an associated standard deviation of 0.15. The sea-based measurements correlate with both the SEVIRI and MODIS AODs to a value of 0.92. Meanwhile the respective satellite retrieval biases with respect to the ship-based measurements are +0.010 for SEVIRI, and +0.029 for MODIS. The root-mean-square differences (RMSD) are 0.059 for SEVIRI and 0.066 for MODIS. Thus the agreement between all three data sources is good, although there is a tendency for the MODIS Dark Target AODs to be positively biased against both the sea-based and the SEVIRI data. The

correlations and RMSDs are broadly in line with previous statistics calculated by B15, however the previous work suggested a slightly higher positive bias by SEVIRI against MODIS, indicating that the average MODIS AODs may have nudged slightly higher (by ~0.01-0.02) from Collection 5 to Collection 6.

A similar picture is present when we look at AERONET (Holben et al., 1998) sun-photometer data taken at KAUST Campus (22.31°N, 39.10°E), here analysed for the years of 2012-2015. This site has also been established and organised by KAUST,

with Georgiy Stenchikov as the PI (http://aeronet.gsfc.nasa.gov/new_web/photo_db/KAUST_Campus.html). Scatterplots of satellite retrieval AODs against AERONET AODs are shown in Figure 6, panels (a) and (c). The Abu Al Bukhoosh site (panels (b) and (d), at 25.50°N, 53.15°E) will be discussed in further detail in Section 6. A caveat for the KAUST site is that we are comparing AERONET measurements taken over land, albeit very close to the shoreline, with satellite retrievals taken over sea. The same spatial matching criteria are applied as for the ship-based measurements, although due to the higher

frequency of observations on land we restrict the temporal matching to within half an hour either side. Level 2 AERONET data are used for KAUST, which is cloud-screened following the technique described by Smirnov et al. (2000). Given the urban-industrial environment at Jeddah $\sim$80 km down the coast from KAUST, we also apply the criterion that the AERONET Ångström coefficient must be less than 0.6, in order to distinguish dust as being the dominant aerosol (e.g., Dubovik et al.,

2002a; Schepanski et al., 2007; Banks and Brindley, 2013). Over the four years considered there are 595 points when all of the three instruments made a simultaneous successful retrieval over KAUST, in which the AERONET measurements had a mean AOD of 0.47±0.34. The SEVIRI retrievals were correlated with AERONET to a value of 0.96, and MODIS was correlated with AERONET to a value of 0.92. Meanwhile the respective biases were 0.000 and +0.080, and the RMSDs were 0.095 and 0.210. The MODIS RMSD is particularly high due to a handful of points at AERONET AODs $> 2$ where MODIS retrieves

AODs $> 4$ (Figure 6(a)), which contributes to MODIS' high positive bias. Hence while the overall picture is consistent with the ship-based measurements, there seem to be higher RMSDs when sea pixels near the coast are considered, perhaps due to the higher mean AODs but perhaps also due to the higher variability in the sea-surface reflectance in coastal waters as opposed to in the open ocean. Another possible factor is the extra distance between the site and the satellite-retrieved pixels given that the retrievals avoid pixels over land and coastal waters.

**5   Inter-comparisons between retrievals**

From Figures 3 and 4 it is apparent that the climatological monthly summertime SEVIRI AOD over the Red Sea often exceeds 1, a range not encapsulated by the ship cruise measurements. It is important to check the agreement of the SEVIRI AODs with other retrievals such as MODIS in this regime, given that the higher the AOD, the larger the impact of dust will be on the radiative energy budget at the surface, within the atmosphere and at the top-of-the-atmosphere. Figure 7 adapts Figure 2 to

compare MODIS AOD retrievals with the coincident SEVIRI retrievals (averaged at 0700, 0800, and 0900 UTC for Terra and at 1000, 1100, and 1200 UTC for Aqua), along with statistics of the inter-retrieval offset and root-mean-square difference. 0800 and 1100 UTC are the approximate daily overpass times for Terra and Aqua over the longitude of the Red Sea, averages of the SEVIRI retrievals are taken either side of the main overpass times in order to provide successful retrievals when sun-glint may prevent SEVIRI retrievals, as is the case for coincident Terra overpasses in the morning in mid-summer. These monthly

statistics are calculated by investigating the coincident and binned 0.125° MODIS and SEVIRI grid cells, when and where both instruments make a successful aerosol retrieval. Note that the original MODIS AOD retrievals at 550 nm have been scaled using the retrieved Ångström coefficient to 630 nm in order to match the wavelength retrieved by SEVIRI.

For almost all months, the MODIS retrievals have higher values than do the SEVIRI retrievals, with the largest values of offsets and RMSDs when the dust loading is largest. Consequently, the offsets and RMSDs tend to be larger in the south of the

Sea than in the north. The correlations are also slightly worse in the south: the average monthly correlation is 0.93 in the north and 0.91 in the south. MODIS-SEVIRI offsets at the basin scale can be much larger than at the ship-point scale, exceeding +0.3 in the southern Red Sea for the Julys of 2009, 2011 and 2013. A caveat to this general pattern is in the autumn to early-spring months when the mean AOD is low, and there are months when there is a very slight negative MODIS offset. Given this

behaviour, the latitudinal gradient in the AODs is also readily apparent in the MODIS-SEVIRI comparison statistics (Figure 7(c,e)). The AOD offsets and RMSDs are quite negligible in autumn and winter (below ±0.05 of offset and below ∼0.1 of RMSD), but show much more variability in July in the south, when there is the most variability in the intensity of large dust outbreaks. Outside of the summer months, retrievals over the southern part of the Sea in April 2015 also show large deviations, 5   a reflection of the relatively high dust loadings seen in this month and location.

The discrepancy between MODIS and SEVIRI at high AODs is systematic, as can be seen in Figure 8, which considers all of the co-located SEVIRI and MODIS retrievals over the eleven-year period. Comparing MODIS against SEVIRI, the correlation coefficient exceeds 0.9 for all months, with the RMSDs and offsets peaking when the AOD is highest. Strikingly, the 2D histogram curves further away from the one-to-one line as the AOD increases, as SEVIRI appears to become less sensitive to 10  the increases in AOD that MODIS seems to observe. Hence in April and July, when the tail of high AODs is extended, the two retrieval datasets diverge the furthest. Very similar patterns are seen in comparisons between Terra-MODIS and SEVIRI, and Aqua-MODIS and SEVIRI (not shown), indicating broad consistency between the separate MODIS instruments.

An implication of this is that there are two regimes for the MODIS/SEVIRI comparisons, which can be seen more precisely if we compare MODIS AODs against the MODIS-SEVIRI offset. At low AODs < 1, the MODIS-SEVIRI offset is small, and 15  neither instrument retrieval is consistently reporting higher AODs than the other. Consequently the correlation between the MODIS AOD and the MODIS-SEVIRI offset in January and October is weak, at 0.47 and 0.41 for the respective months. In spring and summer, however, there is a marked increase in the correlation between MODIS and its offset against SEVIRI, increasing to 0.83 and 0.85 in April and July.

Examining the MODIS retrieved Ångström coefficients over this period provides more detail as to what is occurring here. At 20  low AODs < 1 the mean Ångström coefficient is 0.67, while at AODs > 1 the mean is 0.14. The low value of the latter is a clear signature of desert dust (Dubovik et al., 2002a), given that sea salt alone would not result in such high AODs. Hence it is clear that the substantial majority of the points at the tail end in Figures 8(b) and (c) are composed of desert dust, and so it is dust in some form which is responsible for the high-AOD discrepancy between the SEVIRI and MODIS retrievals. Contamination by other aerosol types is much more likely at lower AODs, when urban/industrial pollution from the major centres along the Red 25  Sea coast (e.g. Jeddah, Port Sudan) may contribute more to the overall signal.

This positive offset behaviour exhibited by MODIS with respect to SEVIRI leads naturally to the question: which retrieval is more accurate at the highest AODs? It is difficult to assess this using data from the ground or from the sea surface given the paucity of measurements made at the surface of such intense events. The ship-board sun-photometers only measured a maximum AOD of 0.86, while the AERONET site at KAUST Campus has only made co-located measurements of AODs > 1 30  for 42 points out of 595 (7%). Within this subset of 'high AOD' data, the MODIS bias with respect to AERONET is +0.313 while the SEVIRI bias is -0.075; this is in comparison with the biases at low dust loadings of +0.063 for MODIS and +0.006 for SEVIRI. Hence as the dust loading increases so does the MODIS bias with respect to the AERONET validation dataset while SEVIRI becomes slightly negatively biased. However, this is a relatively small subset of cases from which it is dangerous to draw any substantiative conclusions. With this in mind we introduce the use of MISR aerosol retrievals to investigate whether 35  they show more coherent behaviour with one or other of the MODIS or SEVIRI set of AOD retrievals. It is important to note

that MISR has quoted uncertainties of equivalent magnitude the MODIS and SEVIRI retrievals (see Section 2), and so they are included here simply for another point of comparison.

As previously reported in comparisons between MISR and Terra MODIS Collection 5 retrievals for all aerosol types over the global ocean for January 2006 (Kahn et al., 2009), MISR retrieves higher AODs than does MODIS when the AOD is low, whereas when the AOD is greater than 0.2-0.3 then the MODIS AODs become 'systematically' higher than the MISR AODs. A similar picture can be found over the Red Sea when we look at scatterplots of MISR/MODIS comparisons for the months of January, April, July and October for the years of 2008-2011 (Figure 9(b)). In this case the datasets have been gridded at 0.25° resolution to take into account the 17.6 km spatial resolution of the MISR aerosol product. These have been co-located with successful SEVIRI retrievals, with the MODIS and SEVIRI AOD datasets reported at, or scaled to, 550 nm, while the MISR AODs used are reported at 558 nm. MODIS and SEVIRI have been retrieved from Aqua or at the Aqua time, due to persistent sun-glint affecting both instrument retrievals along the MISR track at the Terra morning time during the spring and summer months. This persistent sun-glint issue has been noted before by Kahn et al. (2009), in their Figure 4(c), which indicates consistent failure of co-located AOD retrievals between MISR and Terra-MODIS in the northern tropics in July.

The temporal discrepancy accounts for much of the scatter in Figure 9, especially in July. At lower AODs in the winter months there is more temporal stability in the daytime AOD values and hence less scatter. In all four months the overall offset of AOD retrievals from Aqua-MODIS is negative when compared to MISR. However, consistent with Kahn et al. (2009) this masks contrasting behaviour in different AOD regimes which is most obviously manifested in July, the month which has the smallest overall offset. Over the whole dataset, positive offsets at higher AODs (+0.188 when MISR AOD $> 1$) are obscured by negative offsets at lower values (-0.045 when MISR AOD $< 1$).

Taking MISR AOD retrievals as our reference, Figure 9(a) indicates that SEVIRI also has a negative offset as compared with the MISR retrievals but that this offset is consistently in one direction regardless of the MISR AOD. Using a MISR AOD equal to 1 as an arbitrary cut-off between high and low dust loading, SEVIRI has a negative offset of 0.062 (high AOD) and of 0.071 (low AOD) relative to MISR. Thus the discrepancy between regimes is much reduced compared to MODIS. While it is impossible to state which, if any, of the retrievals are correct, it is clear that there are substantial, and in some cases obviously systematic differences between all of the retrievals when elevated dust optical depths are seen over the Red Sea.

There are a number of possible explanations for the discrepancies seen between MODIS and SEVIRI at high AODs. The aerosol model used in the SEVIRI retrievals assumes a purely scattering aerosol. While measurement campaigns indicate that dust typically has a high single-scattering albedo (SSA) in the visible, some absorption is always observed. For example, aircraft measurements during the GERBILS campaign in June 2007 indicate SSA values between 0.92 and 0.99 (Haywood et al., 2011); more recent values derived from Fennec campaign measurements indicate rather lower values, ranging from 0.70 to 0.97 (Ryder et al., 2013). Hence one might anticipate that SEVIRI would retrieve a smaller AOD than that which would be obtained using a model with a lower SSA, all other aspects remaining the same. Similarly, another contributor to the discrepancy may be the different spectral ranges used by the retrievals: while SEVIRI only uses the 630 nm channel for this AOD dataset, the MODIS ocean AODs are retrieved using a combination of six of the MODIS channels between 550 and 2110 nm, which may increase the information content of the MODIS retrieval and hence the MODIS retrieval may be more

sensitive to high loadings of large desert dust particles. A further candidate is the particle sphericity assumptions made by both the MODIS and SEVIRI retrievals. MISR is the exception in this regard, since as was pointed out in Section 2.3, of the eight particle components in the MISR retrieval, two are non-spherical dust analogues. Moreover, the fact that MISR measurements are made at multiple viewing angles allows for increased resolution of the observed aerosol particle scattering, and hence a more constrained knowledge of the phase function. The spherical assumption implies that light scattering occurs due to Mie scattering, which may lead to larger errors at specific scattering angles. Looking again at the AERONET comparisons in Figure 6, the points have been colour-coded by scattering angle between the Sun and the satellite detectors. Note that the colour-coding is different between MODIS and SEVIRI, due to the very different scattering angle ranges observed by each. While SEVIRI reveals no apparent pattern in retrieval quality with respect to scattering angle, at high AERONET AODs the MODIS retrievals show more divergence due to scattering angle, between the high bias of the points at $\sim 100°$ and the low bias of those at $\sim 160°$. Meanwhile the one clear outlier in the MODIS retrievals with respect to the Abu Al Bukhoosh AERONET data (Figure 6(b)) is also within the 96-110° range. Given the limited sample size it is hard to draw too strong a conclusion from this.

To increase the sample size of the scattering angle range, Figure 10 revisualises Figure 8 to learn more about the MODIS and SEVIRI scattering angles with respect to high MODIS and SEVIRI AODs. Looking at April, for SEVIRI AOD retrievals in excess of $\sim 0.5$ there is a systematic tendency for the co-located MODIS AODs to increase with reducing MODIS scattering angle, which may result in the positive MODIS versus SEVIRI offsets seen in Figures 7 and 8 at high AODs. The scattering angle range of 100-120° has previously been diagnosed as a potential source of positive offsets when spherical dust assumptions are used by satellite retrievals (e.g., Wang et al., 2003). Examples of the differences in the phase functions between spherical and non-spherical dust can be seen in Figure 4 of Dubovik et al. (2002b) and Figure 2 of Koepke et al. (2015), both of which indicate that up to a scattering angle of $\sim 80°$ there is very little difference in the phase functions (at 870 and 550 nm, respectively). The phase functions diverge at higher angles, with the spherical phase functions having the smaller values between $\sim 80-130°$, and the larger values at angles $>\sim 150°$. At $\sim 120°$ is the maximum underestimate of the spherical phase function with respect to the non-spherical values, which may lead to a corresponding overestimate of the AOD, especially apparent in Figure 10(a); simultaneously the SEVIRI scattering angles are at their highest (Figure 10(b)), in a range where the phase functions may be overestimated and hence the AODs underestimated. This combination of potentially overestimated MODIS AOD and potentially underestimated SEVIRI AOD may contribute to the diversion between the two AOD datasets in this regime.

The AERONET and MODIS retrievals can provide more information about the aerosol content, for example both retrieve the aerosol Ångström coefficient. For the AERONET comparisons described in Section 4, the mean Ångström coefficients for MODIS and AERONET over KAUST are 0.45 and 0.36, respectively, while over Abu Al Bukhoosh they are 0.51 and 0.35; the greater Ångström coefficients retrieved by MODIS implies that the MODIS retrieval is assuming a smaller aerosol size than AERONET. That the MODIS retrieved Ångström coefficients are persistently high has also been noted by Levy et al. (2003). The correlations are quite weak, at 0.42 and 0.22. Could the possible overestimate of the MODIS Ångström coefficients have an impact on the AOD comparisons, given that the Ångström coefficient is used to scale the MODIS AODs? The MODIS AOD at 550 nm is multiplied by a factor of $(550/630)^{\alpha}$, where $\alpha$ is the Ångström coefficient. If we consider the mean Ångström coefficients at KAUST, if we use the mean MODIS value of 0.45 then this factor is 0.941, if we use the mean AERONET value

of 0.36 then the factor is 0.952: hence the MODIS 'overestimate' of the Ångström coefficient implies an underestimate of the AOD at 630 nm. The Ångström coefficient does not contribute to the elevated MODIS AODs.

Exploring the aerosol size further, the retrievals can also be sub-divided into the aerosol fine and coarse modes, using output from the AERONET Spectral Deconvolution Algorithm (O'Neill et al., 2003), and retrievals of fine and coarse mode MODIS
AODs derived from the retrieved fine-mode fraction (e.g., Kleidman et al., 2005). These comparisons are carried out at 500 nm and are listed in Table 2. Quickly apparent is the strength of the agreement between the coarse-mode retrievals as compared with the fine-mode, indicative of the dominance of the coarse-mode dust to the AOD over these AERONET sites. Strikingly the MODIS biases are greater for the fine-mode than for the coarse-mode despite the fine-mode AERONET AODs being of order 2-3 times less than the coarse-mode AODs. This MODIS overestimate of the fine-mode AODs with respect to the
AERONET SDA algorithm has been noted before by Kleidman et al. (2005), who observed an overestimate of the MODIS fine-mode fraction of the order of 0.1 for dust-dominated conditions, and similarly Levy et al. (2003) noted that the MODIS AOD retrieval has a tendency to underestimate the size distribution. This may also be related to the higher MODIS Ångström coefficients mentioned earlier. The overestimate in the MODIS fine-mode fraction contributes over half of the bias in the total AODs; in contrast, the MODIS and AERONET coarse-mode AODs are more highly correlated with each other, and are a
smaller contributor to the overall bias.

## 6   Contrast with the Persian Gulf

A striking feature of Figure 3 is the contrast in dust loading between the Red Sea and the Persian Gulf (the 'Gulf'), both largely enclosed basins either side of the Arabian Peninsula. The Persian Gulf, also known as the Arabian Gulf, is subject to atmospheric flows and dust storms emanating from Iraq (e.g., Giannakopoulou and Toumi, 2011; Smirnov et al., 2002) and
Saudi Arabia, occasionally also from Iran. Closer to the active source areas, the Gulf is often affected by Middle Eastern dust storms earlier during their life-cycle than is the Red Sea. As with the Red Sea, the Gulf has quiet dust activity during autumn and winter, but there is a high baseline of dust presence throughout the Gulf in spring and summer. This is less intense than in the southern Red Sea in summer, but is a more prolonged feature, and has a more homogeneous spatial pattern within its basin. Due to its closer geographical proximity to the northern Syrian and Iraqi dust sources which flow directly into it, and due to the
wind patterns in this region (Giannakopoulou and Toumi, 2011), the Gulf is more heavily influenced by the spring-time dust events from the Middle East than the Red Sea.

As presented by Figure 7(b), dust activity peaked in the Gulf in spring/summer 2009 and in March 2012. The former period was a period of intense dust activity for Iraq and Iran, especially July 2009: high $PM_{10}$ (Particulate Matter less than $10\,\mu m$ in diameter) concentrations in Tehran were recorded at over $250\,\mu g\,m^{-3}$ during this dusty period (Givehchi et al., 2013).
Meanwhile the latter peak in the Gulf was the month of a large dust event which swept down over Arabia and eventually to the Red Sea, originating in Syria and Iraq. Simulations (Jish Prakash et al., 2015) suggest that this particular event deposited 3.0 Mt of dust to the Arabian Sea, 2.2 Mt to the Gulf, and 1.2 Mt to the Red Sea. This contrast between the Gulf and the Red

Sea is broadly supported by the March 2012 monthly mean AODs, from SEVIRI these are 0.99 over the Gulf and 0.42 over the Red Sea.

How does this contrast affect the satellite retrievals? The SEVIRI/MODIS comparison statistics across the entire Persian Gulf show broad similarity with the Red Sea statistics in Figure 8. Over the Gulf the SEVIRI-MODIS offset at MODIS AODs < 1 is -0.010, at higher AODs it is -0.378; over the Red Sea these are -0.010 and -0.416. The individual monthly statistics for both seas are much the same, except for a greater prevalence of intense dust events in January over the Gulf than over the Red Sea. For January, April, July and October the respective offsets are -0.002, -0.012, -0.040 and -0.0004. What this therefore suggests is that both retrievals' performance over the Gulf are broadly in line with their performance over the Red Sea. This is particularly reassuring for SEVIRI given its geostationary orbit, which is at the limit of its viewing capabilities over the Persian Gulf, with viewing zenith angles >60°.

The Abu Al Bukhoosh AERONET site provides additional information to support this argument, as a maritime site located far from shore on an oil platform in the southern Persian Gulf (25.50°N, 53.15°E), with L2 data available in the 2005-2015 period from November 2006 to September 2008. MODIS and SEVIRI validation statistics and scatterplots reveal a similar picture (Figure 6(b) and (d)) as is present over KAUST, although there are fewer substantial dust events observed. The RMSDs show very minor differences to those seen at KAUST, although it is interesting to note that the biases are somewhat higher: for MODIS these are +0.08 at KAUST and +0.13 at Abu Al Bukhoosh, while for SEVIRI these are +0.00 and +0.08. That both retrievals see this increase in bias suggests some common factor influencing the retrievals, perhaps due to local environmental conditions or due to the nature of the aerosol present. The mean MODIS Ångström coefficient is 0.51, as opposed to 0.45 over KAUST. This implies firstly that MODIS is assuming smaller dust particles over Abu Al Bukhoosh than over KAUST, and that the MODIS AOD-scaling process multiplies the 550 nm AOD by a smaller value than over KAUST.

The difference in aerosol type between the two seas becomes more apparent when we compare the MODIS retrieved Ångström coefficients across the entirety of the two seas. For the eleven years of 2005-2015, over the Red Sea the mean Ångström coefficient is 0.66 (0.67 for low AODs < 1, 0.14 for high AODs > 1, as discussed in Section 5). In contrast, over the Persian Gulf the mean is 0.96, for low AODs it is 0.99 and for high AODs it is 0.26. For clarity, the basin-scale comparisons are not filtered by Ångström coefficient values < 0.6 as are the AERONET comparisons. The Persian Gulf is clearly a more industrial environment than is the Red Sea, as a centre of global oil extraction both on the surrounding land and within the Gulf itself. As a result smaller industrial aerosols are a greater contributor to the aerosol loading over the Persian Gulf than over the Red Sea, as evidenced by higher Ångström coefficients, consistent with previous AERONET measurements of the Ångström coefficient from Bahrain as reported by Dubovik et al. (2002a).

## 7  Conclusions

Satellite retrievals of AOD from SEVIRI over the Red Sea show a clear climatological pattern of high summertime atmospheric dust loading over the southern part of the basin which contrasts with much reduced activity in the north. This pattern induces marked differential radiative heating over the Sea (B15). In this study we have extended the temporal range of the AOD

observations over the Sea to cover the eleven years from 2005-2015 and, importantly, evaluated their performance against an expanded set of 'ground-truth' measurements. In addition, in the light of previously identified biases between different aerosol retrieval algorithms in the presence of dust (e.g., Banks et al., 2013; Carboni et al., 2012), we have compared retrievals from MODIS Collection 6 and MISR with the SEVIRI observations.

There is a high degree of inter-annual variability present in this summertime latitudinal gradient of dust presence over the Red Sea, manifested most clearly in the south in July, where band monthly mean AODs range from ∼0.5 in 2007 to ∼1.8 in 2009. The MACC reanalysis dataset for these two years provides the suggestion that July dust transport over the Sea is dominated by dust sources in Arabia at high-altitude atmospheric layers, whereas near-surface dust tends to be African in origin. Over the past eleven years of the satellite records some patterns in dust activity can be discerned. The increase in dust

activity over the Arabian Peninsula from 2007-2013 (Notaro et al., 2015) can be argued to have had an impact on dust loadings over the Red Sea, but unlike over the neighbouring desert regions to the east this cannot be said to have been a 'regime shift'. The summers of 2008-2013 have been particularly dusty over the southern Red Sea, judging by the SEVIRI retrieved AODs, but recently this has tapered off slightly in 2014 and 2015. Similarly the Persian Gulf has seen a number of increases in dust activity, such as in 2008-2009 and 2012. This is consistent with the analysis of Klingmueller et al. (2016), who identified a

positive dust AOD trend over Saudi Arabia and the wider Middle East from 2000-2012, a trend which has been interrupted by reduced AODs over the last few years. However we also note a recent uptick in dust AODs over both seas in spring 2015.

Over the Red Sea there is broad agreement between SEVIRI, MODIS and the surface datasets, although there has been a marginal increase in MODIS AODs by ∼0.01-0.02 from the previous Collection 5 dataset (B15). The eleven-year satellite retrieval record has enabled the identification of a pronounced positive AOD offset by MODIS with respect to SEVIRI. At lower

AODs of $< 1$, SEVIRI and MODIS are in good agreement, with a MODIS-SEVIRI offset of +0.01. At higher AODs of $> 1$ the retrieved AODs diverge substantially, leading to inter-retrieval offsets of +0.42: at such high loadings the offsets appear to have a pronounced and systematic optical depth dependence. This may be a consequence of the spherical dust assumptions used by the retrievals, leading to variable sensitivity to dust presence at different sun-dust-satellite scattering angles, an issue which merits further investigation in future studies. Leading on from this, a related open question for future studies is whether this

sensitivity is influenced by the origin of the dust, from Africa or from Arabia: particle shape and mineralogy may be different between the two source regions. Comparisons with co-located MISR measurements at high AODs indicate that SEVIRI is offset to a value of -0.06 with respect to MISR, while MODIS is offset to a value of +0.19: these are broadly opposite and similar in magnitude, although SEVIRI does show more consistency between AOD regimes. Meanwhile inter-retrieval comparisons carried out over the Persian Gulf indicate the consistency of the behaviour of the retrievals over a similar enclosed sea affected

by dust outbreaks from its neighbouring deserts. Despite the Gulf's proximity to the far edge of SEVIRI's field-of-view, SEVIRI and MODIS display very similar patterns with respect to each other as they do over the Red Sea, indicating that over both seas the dominant source of any discrepancies between the two retrievals arises simply from the magnitude of the dust activity. Whichever retrieval is most accurate, these differences in behaviour at such high dust loadings will have consequences for the magnitude of the associated differential radiative heating of the surface and atmosphere over the Red Sea, and hence

the atmospheric and oceanic circulation response.

*Acknowledgements.* JRB and HEB have been partially supported for this work by research grant KAUST CRG-1-2012-STE-IMP. GS was supported by the King Abdullah University of Science and Technology. The KAUST Campus AERONET site has been established by KAUST, and GS is the site PI; thanks also to the site managers Jish Prakash and Illia Shevchenko for maintaining the AERONET facility. We thank also the PI and staff of the Abu Al Bukhoosh AERONET site for establishing and maintaining this facility. The ship cruises were

5    also organised by KAUST, within the framework of the Maritime AERONET program, and GS is the PI for the aerosol measurements. The ship cruise data are available from the Maritime AERONET program at http://aeronet.gsfc.nasa.gov/new_web/maritime_aerosol_network.html. We thank also colleagues at the Royal Meteorological Institute of Belgium for the provision of the GERBlike dataset which stores the SEVIRI AOD retrievals, and also for the surface elevation data plotted in Figure 1. MODIS Collection 6 data were provided by the NASA GSFC Level 1 and Atmosphere Archive and Distribution System (available

10   at https://ladsweb.nascom.nasa.gov/data/search.html), and MISR version 22 data were provided by the NASA Atmospheric Science Data Center (http://l0dup05.larc.nasa.gov/MISR/cgi-bin/MISR/main.cgi). We thank the MACC-II project, which has received funding from the EU FP7 under grant agreement number 283576 and was coordinated by the ECMWF for providing the MACC-II reanalysis data set. We would also like to thank Andrew Sayer (at GESTAR/USRA at the NASA GSFC) and another anonymous reviewer for their perceptive comments during the preparation of this manuscript.

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

**Table 1.** Validation statistics between AODs at 630 nm from ship-cruise data and from AERONET at KAUST and at Abu Al Bukhoosh, against SEVIRI and MODIS. Sun-photometer mean AODs include the associated standard deviations. Biases are satellite retrieval AOD - AERONET AOD.

| Comparison | Ship-cruise | KAUST Campus | Abu Al Bukhoosh |
|---|---|---|---|
| Number of points | 255 | 595 | 213 |
| Photometer mean AOD | $0.28 \pm 0.15$ | $0.47 \pm 0.34$ | $0.42 \pm 0.25$ |
| MODIS/Photometer | | | |
| Correlation | 0.92 | 0.92 | 0.93 |
| Bias | +0.029 | +0.080 | +0.134 |
| RMSD | 0.066 | 0.210 | 0.198 |
| SEVIRI/Photometer | | | |
| Correlation | 0.92 | 0.96 | 0.94 |
| Bias | +0.010 | 0.000 | +0.078 |
| RMSD | 0.059 | 0.095 | 0.118 |
| MODIS/SEVIRI | | | |
| Correlation | 0.94 | 0.96 | 0.93 |
| Offset | +0.019 | +0.080 | +0.056 |
| RMSD | 0.056 | 0.194 | 0.150 |

**Table 2.** Comparison statistics between fine- and coarse-mode AERONET and MODIS AODs at 500 nm, at KAUST and at Abu Al Bukhoosh.

|  | KAUST Campus | Abu Al Bukhoosh |
|---|---|---|
| n | 1074 | 773 |
| Total |  |  |
| $\overline{AOD}$ | $0.53 \pm 0.33$ | $0.56 \pm 0.31$ |
| Correlation | 0.93 | 0.92 |
| Bias | +0.097 | +0.191 |
| RMSD | 0.193 | 0.311 |
| Fine-mode |  |  |
| $\overline{AOD}$ | $0.18 \pm 0.08$ | $0.13 \pm 0.06$ |
| Correlation | 0.56 | 0.68 |
| Bias | +0.052 | +0.152 |
| RMSD | 0.110 | 0.183 |
| Coarse-mode |  |  |
| $\overline{AOD}$ | $0.35 \pm 0.29$ | $0.43 \pm 0.26$ |
| Correlation | 0.90 | 0.90 |
| Bias | +0.045 | +0.039 |
| RMSD | 0.156 | 0.203 |

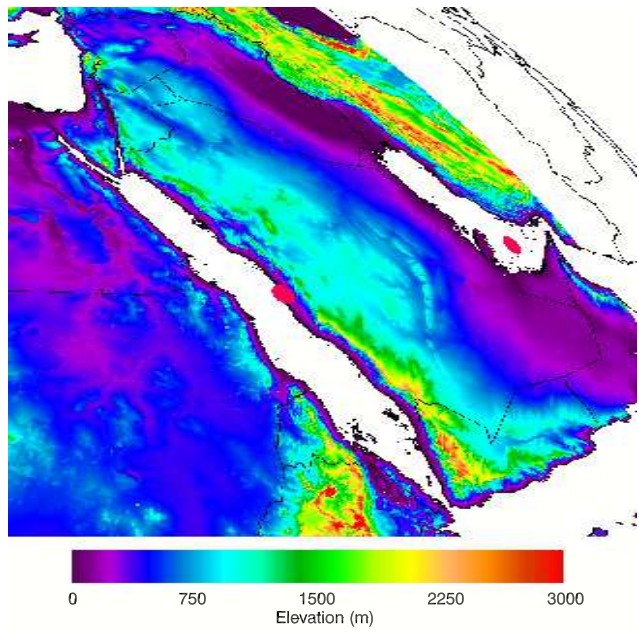

**Figure 1.** Map of the Middle Eastern domain of interest, on the SEVIRI projection: the KAUST AERONET site on the Red Sea coast is marked as a red spot, as is the Abu Al Bukhoosh site in the Persian Gulf. The colour contours represent the surface elevation (as developed by the Eumetsat Satellite Application Facility for Nowcasting (MétéoFrance, 2013)). Elevation data are truncated at the $70°$ viewing zenith angle contour. Note also the increasing curvature of the SEVIRI projection towards the north-east of the field-of-view.

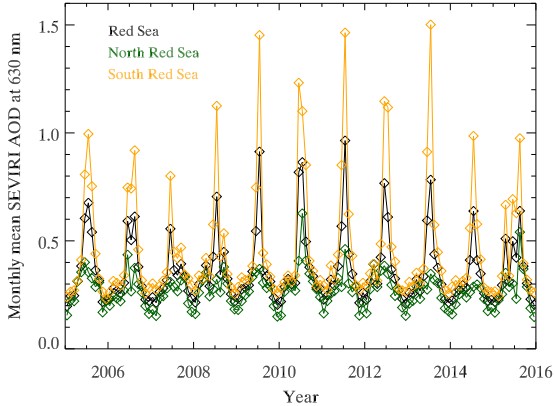

**Figure 2.** Timeseries of monthly average AOD at 630 nm from SEVIRI in the Red Sea. Colours represent individual years, and 'South' and 'North' are divided by the line of $20°$ N.

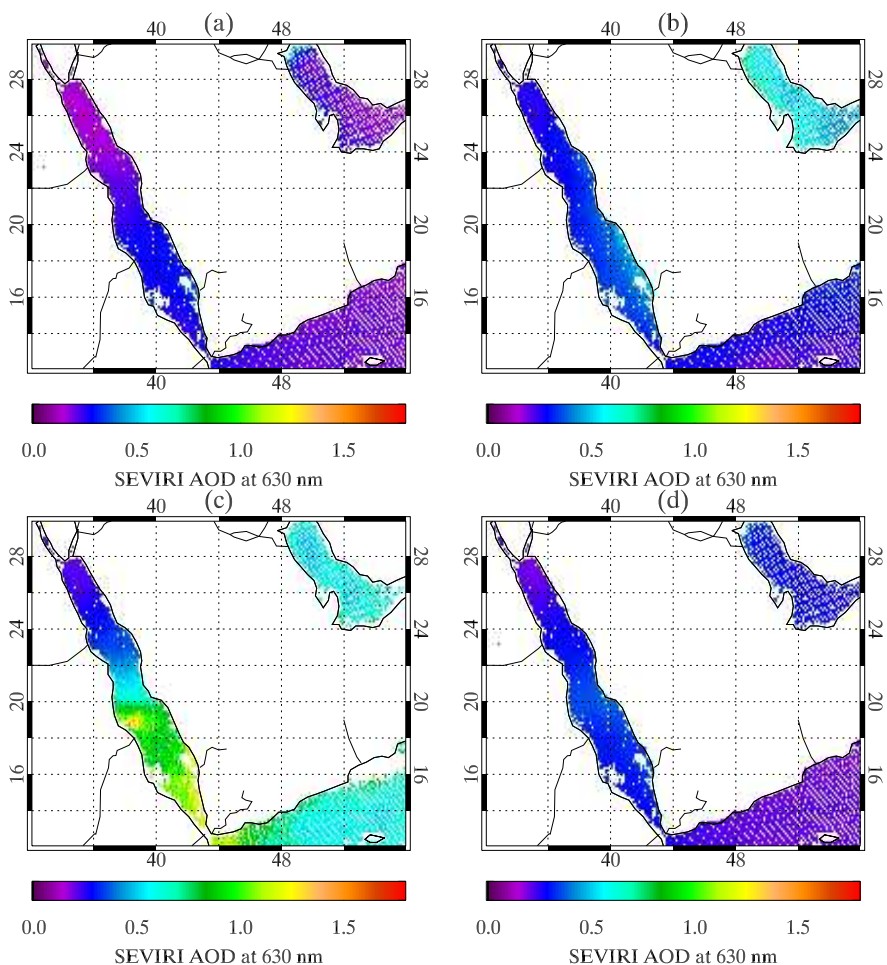

**Figure 3.** Maps of 'climatological' (2005-2015) monthly average AOD from SEVIRI at 630 nm. (a) January, (b) April, (c) July, (d) October.

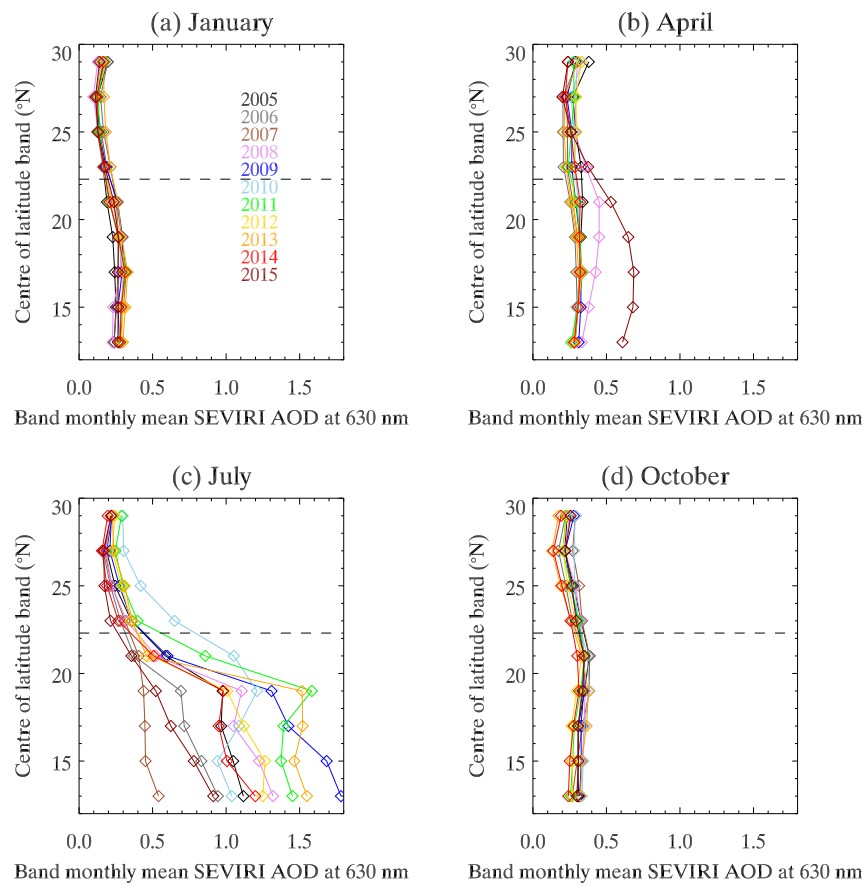

**Figure 4.** Monthly average AOD from SEVIRI by latitude band in the Red Sea. Colours represent individual years, and the dashed line at 22.3°N is the latitude of KAUST.

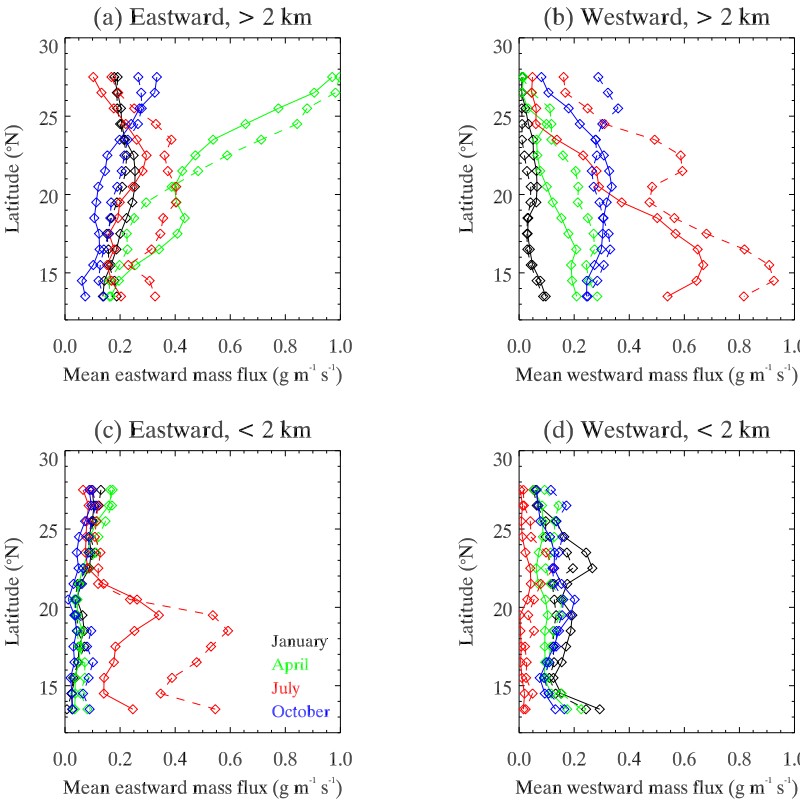

**Figure 5.** Monthly mean dust mass transport by 1° latitude band in the Red Sea, integrated over segments of the atmospheric column. Colours represent individual months, the solid lines represent 2007, and the dashed lines represent 2009. (a) Eastward dust flux, from Africa, between 2-15 km altitude; (b) westward dust flux, from Arabia, 2-15 km; (c) eastward dust flux, between 0-2 km altitude; (d) westward dust flux, 0-2 km.

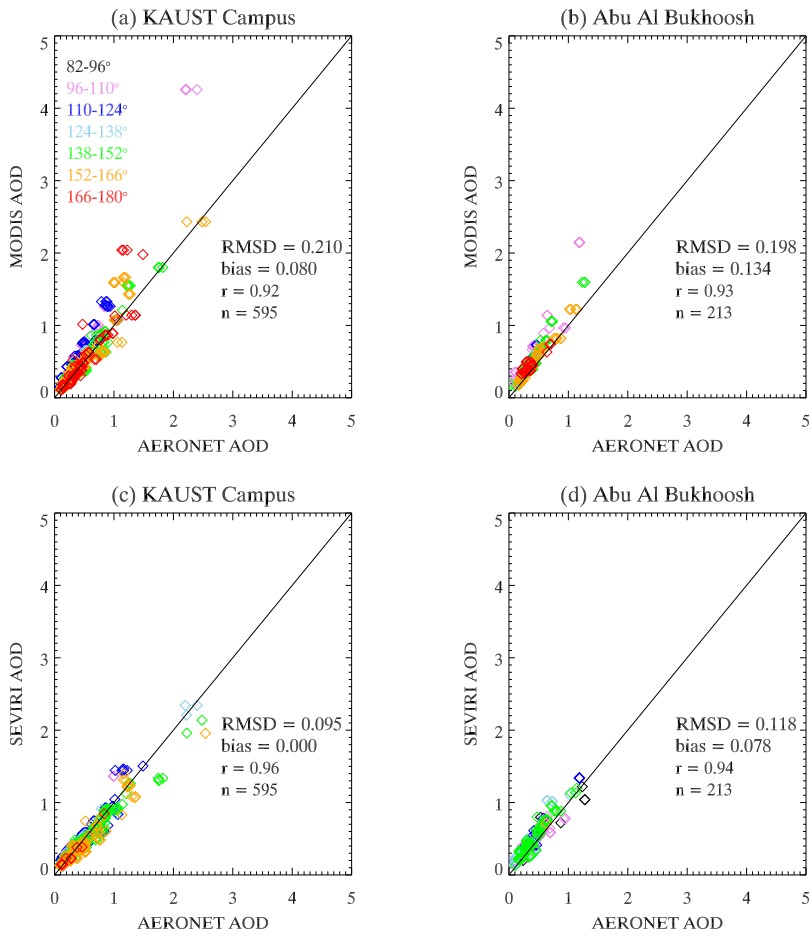

**Figure 6.** Satellite retrieval comparisons against L2 AERONET AODs measured at KAUST Campus, 2012-2015, and Abu Al Bukhoosh, 2006-2008, at a wavelength of 630 nm. Colours represent scattering angles to the satellite instruments.

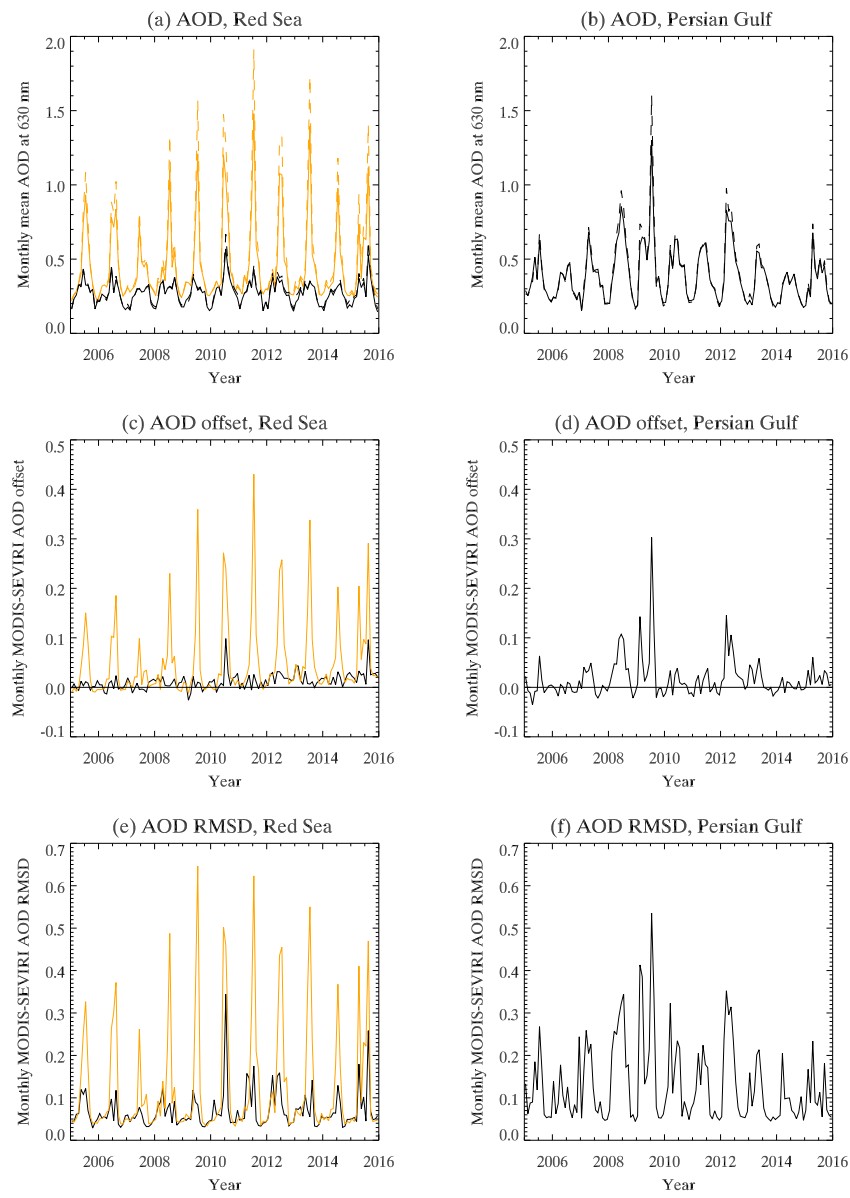

**Figure 7.** Timeseries of monthly average MODIS/SEVIRI AOD statistics over the Red Sea (left column, a, c, e) and the Persian Gulf (right column, b, d, f). (a, b) average AOD at 630 nm: solid lines represent SEVIRI, dashed lines represent MODIS, in the Red Sea black indicates the northern half of the Sea and orange represents the south. (c, d) MODIS-SEVIRI offset. (e, f) MODIS-SEVIRI RMSD.

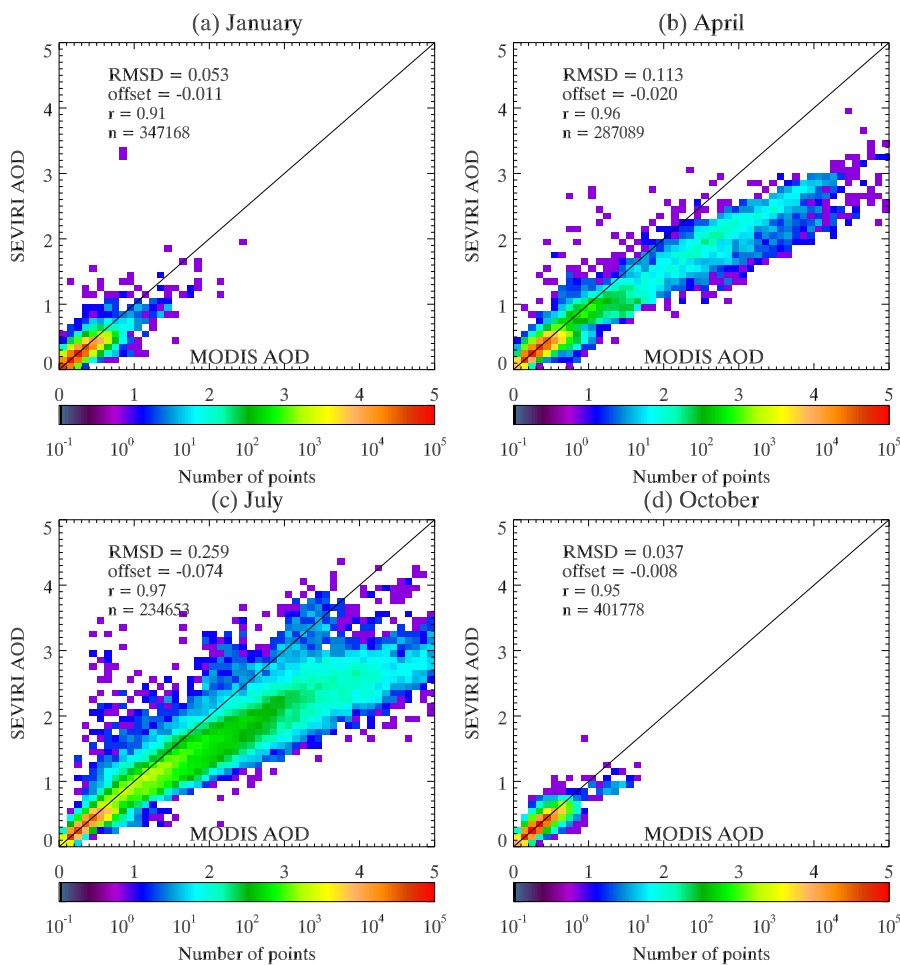

**Figure 8.** Density plots of MODIS AOD against SEVIRI AOD over the Red Sea, 2005-2015.

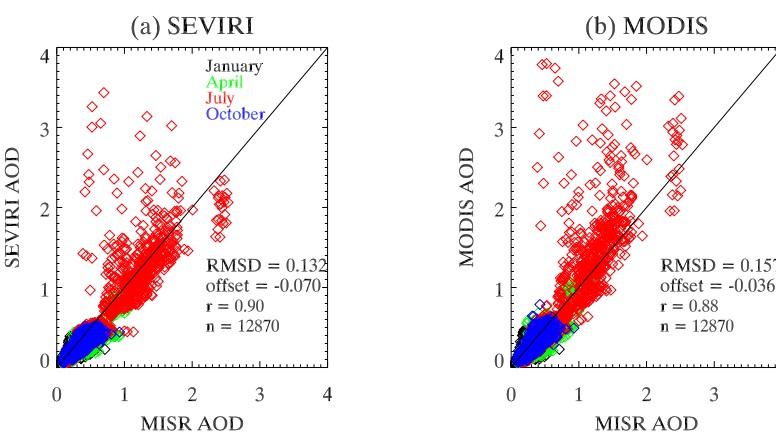

**Figure 9.** Scatterplots of MISR AOD against SEVIRI and MODIS AODs over the Red Sea, 2008-2011. Points are colour-coded by month.

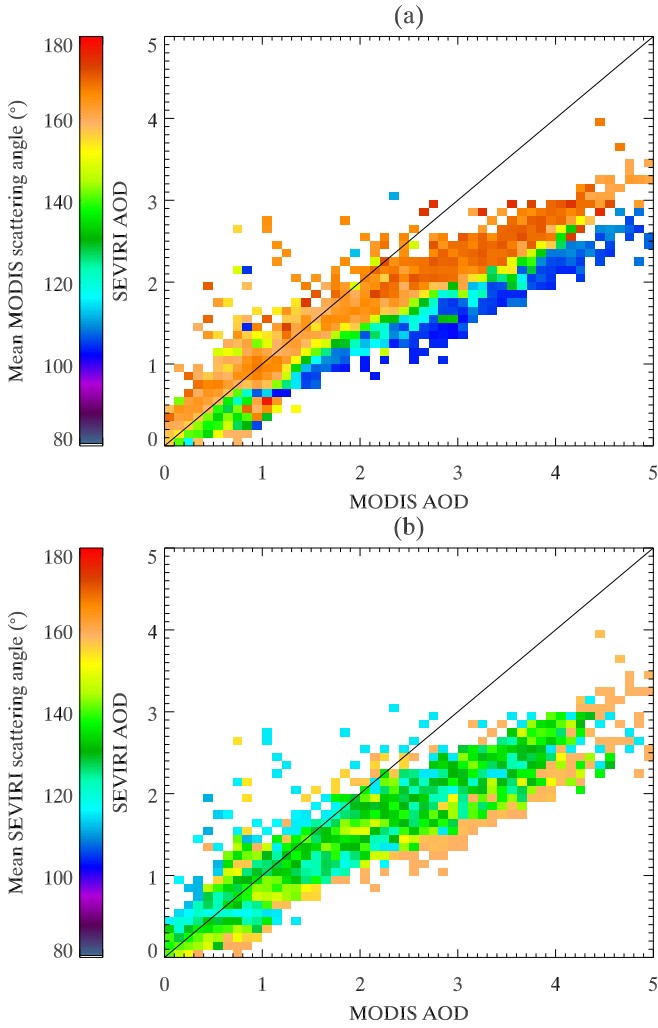

**Figure 10.** MODIS AOD against SEVIRI AOD over the Red Sea during April, 2005-2015, colour-coded by (a) mean MODIS scattering angle, (b) mean SEVIRI scattering angle.