# Peer review of "Satellite retrievals of dust aerosol over the Red Sea and the Persian Gulf, 2005-2015"

_Atmospheric Chemistry and Physics, 2016_

## Referee Comment (RC1) · A. M. Sayer (Referee) · 16 Nov 2016

I am posting this under my name (Andrew Sayer) as I have collaborated with the authors, and the lead author and I were PhD students together. I also provided some comments and suggestions to the authors while the manuscript was in preparation. I feel I am able to provide an objective review.

[Figure]

**Summary**

This paper examines dust aerosol property retrievals over the Red Sea and Persian Gulf over the period 2005-2015 from SEVIRI and MODIS. This is a validation exercise of the two vs. AERONET and MAN, a comparison of the two against each other, and an examination of the seasonal and interannual variability of aerosol loading here. MISR data are also briefly used. It is relevant to the scope of ACP and of interest to the broader scientific community. Some of the work is an update of Brindley et al (2015) although this uses an updated MODIS data version (Collection 6) and has a broader scope so this is a worthwhile extension. In terms of MODIS, this is of particularly utility because there hasn't been much attention in the literature to validation of the MODIS over-ocean AOD product in dusty scenes. In these cases the assumption of aerosol sphericity causes problems in the retrieved products; this has been known about for a long time but not been published about so much. So this analysis highlights that nicely, and may bring more attention to the topic. Levy et al (2003) showed a bit of this, briefly, but for an older version of the MODIS product.

My recommendation is that this paper can be accepted for publication in ACP, after attention to the follow comments. I also have some general suggestions for straightforward extensions to the analysis, which may improve on the utility of the study for the part of the community interested in AOD retrieval development/validation. I don't feel that all of these extensions are necessary for publication but would encourage the authors to consider adding them. I am happy to review a revised version if the editor would like.

**Specific comments**

Title/abstract: I suggest adding the Persian Gulf into here as well, since this isn't mentioned at present in either title or abstract. I'd also mention in the abstract that some differences may be related to sphericity assumptions since, as I noted above, this has not got as much attention as it perhaps should have.

General: correlation coefficients are often given to 3 decimal places; I think 2 is sufficient as 3 gives a false impression of the level of precision on an estimate of correlation.

Page 4, lines 4-5: It is important to make the distinction here that the Dark Target algorithm is **NOT** applied over land and ocean. The product is often colloquially referred to as 'Dark Target' but there is not one Dark Target algorithm. There is one algorithm over land and another algorithm over ocean. They use different wavelengths, make different aerosol assumptions, and different surface assumptions, and have very little in common. This is a common misconception in the literature and is misleading so it would be good to make the text clear. The MODIS algorithm applied in this study is the ocean algorithm. So when Deep Blue is considered as well, this makes three algorithms in the MODIS MxD04 product, not two as line 4 states.

Page 4, line 21: I recommend using pixels with QA=1,2,3, rather than only QA=3. The QA=3 recommendation is made for Dark Target land retrievals. For ocean algorithm retrievals, the team recommend that QA=0 are discarded but QA=1,2,3 can be used. This is because the QA flags were not found to mean much over the ocean (aside from QA=0), is what prior ocean validation exercises were done with, and what is done when aggregating into the widely-used Level 3 products. Doing this will give a more realistic representation of how the product is intended to be used, and should increase data volume as well. I'm not sure whether this will quantitatively affect the conclusions

much or not.

Section 2.2: Does the study use both Terra and Aqua data? I did not see it stated, perhaps I missed it. If both are used, is there any significant difference between the two found?

Page 6 line 25: This says that the assimilation makes the model AOD fields consistent with MODIS. The implication of that wording to me is that model AOD becomes replaced with MODIS AOD, but I assume that this is not the case. I think that this last part of the sentence should be deleted since it might cause confusion and the sentence already says it is 4D-Var and gives a reference.

Page 7 lines 30-32: I would state again that quality flags are applied here, just to be clear to the reader, as sometimes people don't use them (which is bad). I realise it was stated earlier in the paper that they were used for MODIS.

Page 8 lines 14-15, 21: Smirnov et al (2000) should probably also be cited at one of these points as it describes the cloud screening for the level 2 data used.

Page 9, lines 27-30: It's possible that this is due in part to SSA assumptions, since SSA influences the point at which the reflectance vs. AOD curve flattens out. I'm not sure what the difference between MODIS and SEVIRI assumptions is. MODIS ocean retrievals include a green band so there will be some dust absorption at that wavelength, but the SEVIRI wavelengths (and longer MODIS wavelengths) should all be fairly nonabsorbing for dust. (Note that the MODIS ocean product also provides output AOD at 470 nm, where dust absorption would be stronger, but this band is not actually used for the inversion-a few papers are incorrect on that matter.)

Page 10 lines 18-19: it is good to include MISR as another point of comparison, but worth mentioning more directly that as this is another retrieval, with stated uncertainties comparable to MODIS, better agreement with MISR does not necessarily mean that a product is closer to the truth.

General, figures: some look a bit overly digitised, not sure if this is a problem in the source files or the conversion for embedding in the manuscript but this should be checked.

General, figures: most font sizes could do with being increased 1-2 points as a few scales are hard to read at 100 % magnification (this could be related to the above point as well).

Figure 7: I would change the caption to read *MODIS-SEVIRI offset* rather than *MODIS-SEVIRI bias* as people often read bias to imply that the reference is truth, while offset is a more neutral word. Same with some of the text on e.g. pages 10-11.

Figures 7,9: it would be good to include a short plot title rather than just a letter, for easier quick reference to what is in the plot when browsing between text and figures.

**Suggestions for extension**

Both MODIS and SEVIRI over-ocean products retrieve AOD and Ångström exponent; MODIS also retrieves fine-mode AOD fraction (from which one can get fine mode AOD and coarse mode AOD). I'm not sure if it is possible to derive fine mode fraction from

the SEVIRI product as well (my impression from the cited references is that it's not a direct output). It would be interesting to compare the Ångström exponent retrievals with AERONET and with each other, as well, since some of the effects of the MODIS spherical assumption might also manifest there (in the same way as they are seen in the AOD comparisons). This is discussed briefly on page 10 but it does not go into much detail.

The MODIS product could also be broken down to give fine and coarse mode AOD, which could be compared with that from AERONET's SDA product. This product has also not been evaluated widely for MODIS (Kleidman et al., 2005 is the main exception), and not at all to my knowledge for Collection 6. SDA is available at Level 2 for the Kaust site; it is only Level 1.5 for Abu Al Bukhoosh (because I think of the limited number of spectral channels) but that might be able to be used with caveats. Adding this to the analysis would give important information about the quality of this product, as well as perhaps indicating how the 'spherical dust' assumption affects the robustness of total AOD vs. the fine/coarse-mode split.

MISR also has a 'nonspherical AOD' product which could be taken as a proxy for mineral dust AOD, and compared with MODIS-derived coarse-mode AOD in a similar way to e.g. Figure 9.

Is it possible to evaluate the SEVIRI retrievals as a function of time of day (maybe comparing diurnal cycles with AERONET or just plotting bias vs. hour or something)? I understand if there might not be enough data. However temporal resolution is something that geostationary sensors have an advantage over polar-orbiters with, so I think it would be good to emphasise this aspect of the SEVIRI retrieval.

**References**

Kleidman, R. G., N. T. O'Neill, L. A. Remer, Y. J. Kaufman, T. F. Eck, D. Tanré, O. Dubovik, and B. N. Holben (2005), Comparison of Moderate Resolution Imaging Spectroradiometer (MODIS) and Aerosol Robotic Network (AERONET) remote-sensing retrievals of aerosol fine mode fraction over ocean, J. Geophys. Res., 110, D22205, doi:10.1029/2005JD005760.

Levy, R. C., L. A. Remer, D. Tanré, Y. J. Kaufman, C. Ichoku, B. N. Holben, J. M. Livingston, P. B. Russell, and H. Maring (2003), Evaluation of the Moderate-Resolution Imaging Spectroradiometer (MODIS) retrievals of dust aerosol over the ocean during PRIDE, J. Geophys. Res., 108(D19), 8594, doi:10.1029/2002JD002460.

Smirnov, A., B. N. Holben, T. F. Eck, O. Dubovik, and I. Slutsker (2000), Cloud-screening and quality control algorithms for the AERONET database, Remote Sens. Environ.,73 (3), 337-349.

---

## Referee Comment (RC2) · Anonymous Referee #2 · 14 Dec 2016

This paper analyses the inter-annual variability of dust aerosol over the Red Sea with respect to the summer time latitudinal gradient in dust loading using satellite based measurements of aerosol optical depth (AOD). AOD products from both the geostationary SEVIRI instrument and the polar orbiting MODIS instrument are used over a period ranging from 2005–2015. This represents a significant extension in the time period used previously in a similar study from 2008–2012 (Brindley et al., 2015). In addition, this previous study used MODIS Collection 5 whereas the current study uses MODIS collection 6. The SEVIRI and MODIS AOD retrievals are validated against ship based measurements and AERONET measurements made at the KAUST AERONET site. MISR based AOD retrievals are used as another point of comparison. Finally, the performance of the retrieval over the Red Sea is compared to that over the Persian Gulf.

[Figure]

I recommend that the paper be accepted for publication after considering some minor comments that I've given below.

- **Page 4, line 4**: The MODIS 'Dark Target' algorithm typically refers to the dark target algorithm over land not ocean. It is important to make it clear that the retrieval over ocean is different and distinct to the dark target retrieval over land.

- **Page 4, line 8**: MODIS is 250 m, 512 m and 1 km (depending on band), not 10 km.

- **Page 10, line 8**: The SEVIRI retrieval uses measurements over a smaller spectral range (630–1610 nm) than that of the MODIS retrieval (550–2110 nm). Is it possible that the larger wavelength at 2110 nm provides additional sensitivity to large dust particles for the MODIS retrieval compared to the SEVIRI retrieval? Likewise the measurements used for the MISR retrieval have even a smaller maximum wavelength and likewise the MODIS AODs are positively biased relative to the MISR AODs.

- **Page 11, line 4**: Maybe say "Taking MISR retrieved AODs" or in some other way make it clear that MISR is still just another retrieval and should not be taken as truth.

- **Page 11, line 10**: In addition to the non-spherical dust analogues present in the MISR retrieval, as pointed out by the author, MISR makes measurements at multiple view angles which will help resolve a larger range of the single scattering phase function than possible with either SEVIRI or MODIS. This should be taken into account if further discussion of the implication of the sphericity assumption is added.

- **Page 11, line 10**: It may be instructive to compare plots of phase functions for dust particles with and without the spherical assumption for this discussion.

- **Page 19, figure 1, line 3**: A new sentence should start between "contour" and "note".

- The tendency for the MODIS AODs to be positively biased against both the ship borne measurements and SEVIRI AODs may have something to do with the Ångtröm exponent due to its use to scale the AOD at 550 nm to that at 675 nm and 630 nm, respectively. It may be useful to compare the MODIS Ångtröm exponent to the AERONET retrieved exponent at the KAUST AERONET site or to that of the SEVIRI retrieval.

- It might be instructive to compare the Ångtröm exponent of dust over the Red Sea from east and west sources. Likewise, it may be useful to do the same thing when comparing dust over the Red Sea with that over the Persian Gulf.

- The fonts on some of the plots are a bit small and should be increased in size.

---

## Author Comment (AC1) · 3 Feb 2017

We thank Andrew Sayer for his helpful comments. Below are our responses to his comments, and a list of relevant changes.

1) "Title/abstract: I suggest adding the Persian Gulf into here as well, since this isn't mentioned at present in either title or abstract. I'd also mention in the abstract that some differences may be related to sphericity assumptions since, as I noted above, this has not got as much attention as it perhaps should have."

We agree that the analysis over the Persian Gulf is a significant contribution to the paper, so we have added the Persian Gulf to the title. Moreover we have also included the Abu Al Bukhoosh comparison statistics in Table 1. The abstract now includes

a sentence on the consequences of the use of particle sphericity assumptions and the differences in scattering angles observed by the two satellite instruments. In the process of considering the review process we have noticed that our calculations of the SEVIRI scattering angle were not consistent with how it is calculated by MODIS: hence we now include a second panel in Figure 10 to consider the mean scattering angle in relation to the MODIS/SEVIRI AODs, which expands the analysis and provides more detail to the overall picture.

2) "General: correlation coefficients are often given to 3 decimal places; I think 2 is sufficient as 3 gives a false impression of the level of precision on an estimate of correlation."

Correlation values in various Figures and in Table 1 are now only given to two decimal places, rather than three. This is also the case in the text.

3) "Page 4, lines 4-5: It is important to make the distinction here that the Dark Target algorithm is NOT applied over land and ocean. The product is often colloquially referred to as 'Dark Target' but there is not one Dark Target algorithm. There is one algorithm over land and another algorithm over ocean. They use different wavelengths, make different aerosol assumptions, and different surface assumptions, and have very little in common. This is a common misconception in the literature and is misleading so it would be good to make the text clear. The MODIS algorithm applied in this study is the ocean algorithm. So when Deep Blue is considered as well, this makes three algorithms in the MODIS MxD04 product, not two as line 4 states."

This is a very relevant point, that there are two 'Dark Target' algorithms: we have amended the first paragraph of Section 2.2 to clarify this issue, emphasising that we are considering only the product over ocean.

4) "Page 4, line 21: I recommend using pixels with QA=1,2,3, rather than only QA=3. The QA=3 recommendation is made for Dark Target land retrievals. For ocean algorithm retrievals, the team recommend that QA=0 are discarded but QA=1,2,3 can be used. This is because the QA flags were not found to mean much over the ocean (aside from QA=0), is what prior ocean validation exercises were done with, and what is done when aggregating into the widely-used Level 3 products. Doing this will give a more realistic representation of how the product is intended to be used, and should increase data volume as well. I'm not sure whether this will quantitatively affect the conclusions much or not."

We have changed our analysis to include MODIS pixels with QA values of 1 and 2, as well as 3. Quantitatively it makes very little difference, with a relatively small number of extra pixels now included: for example, over the KAUST AERONET site there are now 595 co-locations, compared to 575 beforehand. Correspondingly there has been a very minor adjustment to many of the comparison statistics throughout the manuscript.

5) "Section 2.2: Does the study use both Terra and Aqua data? I did not see it stated, perhaps I missed it. If both are used, is there any significant difference between the two found?"

The first paragraph of Section 5 indicates that we are using both Terra and Aqua data, for clarity we now also write much earlier in the second paragraph of Section 2.2 that we are using AOD data 'from both the Terra and Aqua satellites'. Few significant differences between the satellites are found: density plots of Terra MODIS vs. SEVIRI and Aqua MODIS vs. SEVIRI (analogous to Figure 8) show much the same behaviour between Terra and Aqua MODIS. For July across the Red Sea, the Terra/Aqua correlations with SEVIRI are 0.96/0.97, the offsets are -0.068/-0.080, and the RMSDs are 0.266/0.252. However we concede that there may be smaller-scale differences of greater magnitude which may not be apparent in this analysis. In Section 5 in the first paragraph where the density plots are introduced, we include the sentence: 'Very

similar patterns are seen in comparisons between Terra-MODIS and SEVIRI, and Aqua-MODIS and SEVIRI (not shown).'

6) "Page 6 line 25: This says that the assimilation makes the model AOD fields consistent with MODIS. The implication of that wording to me is that model AOD becomes replaced with MODIS AOD, but I assume that this is not the case. I think that this last part of the sentence should be deleted since it might cause confusion and the sentence already says it is 4D-Var and gives a reference."

We agree that the wording describing the MACC assimilation process is ambiguous and potentially confusing: the last part of this sentence on line 25 has now been removed.

7) "Page 7 lines 30-32: I would state again that quality flags are applied here, just to be clear to the reader, as sometimes people don't use them (which is bad). I realise it was stated earlier in the paper that they were used for MODIS."

An extra statement has been included in the first paragraph of Section 4 to point out that we are using MODIS QA values between 1 and 3.

8) "Page 8 lines 14-15, 21: Smirnov et al (2000) should probably also be cited at one of these points as it describes the cloud screening for the level 2 data used."

We have included the Smirnov et al. (2000) reference in this paragraph, when we declare that we are using L2 data for KAUST.

9) "Page 9, lines 27-30: It's possible that this is due in part to SSA assumptions, since SSA influences the point at which the reflectance vs. AOD curve flattens out. I'm not sure what the difference between MODIS and SEVIRI assumptions is. MODIS

ocean retrievals include a green band so there will be some dust absorption at that wavelength, but the SEVIRI wavelengths (and longer MODIS wavelengths) should all be fairly nonabsorbing for dust. (Note that the MODIS ocean product also provides output AOD at 470 nm, where dust absorption would be stronger, but this band is not actually used for the inversion- a few papers are incorrect on that matter.)"

This is a good point, and we include some sentences to this effect considering this point on page 11 when we discuss possible explanations for the discrepancies between the SEVIRI and MODIS retrievals.

10) "Page 10 lines 18-19: it is good to include MISR as another point of comparison, but worth mentioning more directly that as this is another retrieval, with stated uncertainties comparable to MODIS, better agreement with MISR does not necessarily mean that a product is closer to the truth."

Where we introduce the MISR retrievals in Section 5 we include a sentence pointing out the similar quoted uncertainties of the MISR retrievals compared to the MODIS and SEVIRI retrievals, as a reminder to the reader that MISR should not be regarded as the 'truth'.

11) "General, figures: some look a bit overly digitised, not sure if this is a problem in the source files or the conversion for embedding in the manuscript but this should be checked."

The reviewer appears to be referring to Figures 8 and 10, the density plots of MODIS against SEVIRI AODs. We now use a slightly different routine to create these plots, which we believe now provides more clarity.

12) "General, figures: most font sizes could do with being increased 1-2 points as a

few scales are hard to read at 100% magnification (this could be related to the above point as well)."

We have increased some of the font sizes a couple of points, and we have given Figures 3 and 5 more page space.

13) "Figure 7: I would change the caption to read MODIS-SEVIRI offset rather than MODIS-SEVIRI bias as people often read bias to imply that the reference is truth, while offset is a more neutral word. Same with some of the text on e.g. pages 10-11."

We have amended the captions and labels in Figures 7-9 to say 'offset' instead of 'bias', and at numerous points in the text we have changed occurrences of bias to offset when we are referring to differences between the satellite retrievals.

14) "Figures 7,9: it would be good to include a short plot title rather than just a letter, for easier quick reference to what is in the plot when browsing between text and figures."

Extra information has been added to the plot titles, for ease of reference.

15) "Suggestions for extension. Both MODIS and SEVIRI over-ocean products retrieve AOD and Ångström exponent; MODIS also retrieves fine-mode AOD fraction (from which one can get fine mode AOD and coarse mode AOD). I'm not sure if it is possible to derive fine mode fraction from the SEVIRI product as well (my impression from the cited references is that it's not a direct output). It would be interesting to compare the Ångströexponent retrievals with AERONET and with each other, as well, since some of the effects of the MODIS spherical assumption might also manifest there (in the same way as they are seen in the AOD comparisons). This is discussed briefly on page 10 but it does not go into much detail. The MODIS product could also be broken down to give fine and coarse mode AOD, which could be compared with that from AERONET's

[Figure]

SDA product. This product has also not been evaluated widely for MODIS (Kleidman et al., 2005 is the main exception), and not at all to my knowledge for Collection 6. SDA is available at Level 2 for the Kaust site; it is only Level 1.5 for Abu Al Bukhoosh (because I think of the limited number of spectral channels) but that might be able to be used with caveats. Adding this to the analysis would give important information about the quality of this product, as well as perhaps indicating how the 'spherical dust' assumption affects the robustness of total AOD vs. the fine/coarse-mode split. MISR also has a 'nonspherical AOD' product which could be taken as a proxy for mineral dust AOD, and compared with MODIS-derived coarse-mode AOD in a similar way to e.g. Figure 9. Is it possible to evaluate the SEVIRI retrievals as a function of time of day (maybe comparing diurnal cycles with AERONET or just plotting bias vs. hour or something)? I understand if there might not be enough data. However temporal resolution is something that geostationary sensors have an advantage over polar-orbiters with, so I think it would be good to emphasise this aspect of the SEVIRI retrieval."

We thank the reviewer for his suggestions for extensions, and we agree that including some of this analysis adds extra value to the paper. Comparisons of MODIS and AERONET and Ångström coefficients are detailed at the end of Section 5. SEVIRI is not included in this analysis due to the fact that its Ångström coefficient derivation comes from the relationship between two independent AOD retrievals at different wavelengths, and is not itself a validated product. Because of this, we carry out all AERONET comparisons at 630 nm, scaling the MODIS and AERONET retrievals using their retrieved Ångström coefficients. The MODIS Ångström coefficients have a tendency to be biased high against AERONET, by +0.09 at KAUST and +0.16 at Abu Al Bukhoosh, implying that the MODIS retrieval assumes smaller particles than does the AERONET retrieval, and which coheres with the picture presented by the fine-/coarse-mode MODIS and AERONET comparisons.

Comparisons of MODIS and AERONET fine/coarse-mode AODs have been included at the end of Section 5, with statistics in Table 2. Consistent with Kleidman et al. (2005)

we find that MODIS-AERONET biases are actually greater for the fine-mode than for the coarse-mode, despite the coarse-mode having the greater AOD.

We like the idea of investigating the diurnal cycle of the quality of the SEVIRI retrievals, and have explored briefly the daytime cycle in the SEVIRI-AERONET biases. At KAUST the hourly-resolved biases are at a minimum at the ends of the day (-0.05 between 1300-1400 UTC, and at a maximum of +0.04 between 0800-0900 UTC). The values are however very small, so it is difficult to draw too strong a conclusion from this. Hence while this is an intriguing idea, we do not include it in the final paper.

————————————————————

---

## Author Comment (AC2) · 3 Feb 2017

We thank the reviewer for their helpful comments, and we include our responses below:

1) "Page 4, line 4: The MODIS 'Dark Target' algorithm typically refers to the dark target algorithm over land not ocean. It is important to make it clear that the retrieval over ocean is different and distinct to the dark target retrieval over land."

As with point 3 made by Andrew Sayer, we agree that this is a very relevant point. We have amended the text accordingly.

2) "Page 4, line 8: MODIS is 250 m, 512 m and 1 km (depending on band), not 10 km."

[Figure]

It is true that the MODIS measurements are made at spatial resolutions of 250 m, 512 m, and 1 km, and this is stated in the text. It is the AOD derived products which have a resolution of 10 km: for clarity we have included the statement that this is the resolution of the AOD data within the L2 products.

3) "Page 10, line 8: The SEVIRI retrieval uses measurements over a smaller spectral range (630–1610 nm) than that of the MODIS retrieval (550–2110 nm). Is it possible that the larger wavelength at 2110 nm provides additional sensitivity to large dust particles for the MODIS retrieval compared to the SEVIRI retrieval? Likewise the measurements used for the MISR retrieval have even a smaller maximum wavelength and likewise the MODIS AODs are positively biased relative to the MISR AODs."

We agree that the spectral ranges are an important point when considering the capabilites of the retrievals, and to the discussion on page 11 (Section 5) on possible explanations for the SEVIRI/MODIS discrepancy we add the sentence: "Similarly, another contributor to the discrepancy may be the different spectral ranges used by the retrievals: while SEVIRI only uses the 630 nm channel for this AOD dataset, the MODIS ocean AODs are retrieved using a combination of six of the MODIS channels between 550 and 2110 nm, which may increase the information content of the MODIS retrieval and hence the MODIS retrieval may be more sensitive to high loadings of large desert dust particles."

4) "Page 11, line 4: Maybe say "Taking MISR retrieved AODs" or in some other way make it clear that MISR is still just another retrieval and should not be taken as truth."

'Taking MISR AOD retrievals as our reference' is now our introduction to this paragraph (was Page 11, line 4), to remind the reader that MISR AODs are also retrievals and not the absolute truth.

5) "Page 11, line 10: In addition to the non-spherical dust analogues present in the MISR retrieval, as pointed out by the author, MISR makes measurements at multiple view angles which will help resolve a larger range of the single scattering phase function than possible with either SEVIRI or MODIS. This should be taken into account if further discussion of the implication of the sphericity assumption is added."

Indeed, MISR's observations at multiple viewing angles should help provide the MISR retrieval with more information about the aerosol particle scattering. We now include in this paragraph the sentence: "Moreover, the fact that MISR measurements are made at multiple viewing angles allows for increased resolution of the observed aerosol particle scattering, and hence a more constrained knowledge of the phase function."

6) "Page 11, line 10: It may be instructive to compare plots of phase functions for dust particles with and without the spherical assumption for this discussion."

In the third paragraph from the end of Section 5, where we discuss Figure 10, we also describe the spherical and non-spherical dust phase functions analysed by three previous authors.

7) "Page 19, figure 1, line 3: A new sentence should start between "contour" and "note"."

There is a full stop between 'contour' and 'Note'.

8) "The tendency for the MODIS AODs to be positively biased against both the ship borne measurements and SEVIRI AODs may have something to do with the Ångström exponent due to its use to scale the AOD at 550 nm to that at 675 nm and 630 nm, respectively. It may be useful to compare the MODIS Ångström exponent to the AERONET retrieved exponent at the KAUST AERONET site or to that of the SEVIRI

retrieval."

It is worthwhile to explore the Ångström coefficient in more detail, we agree, and so as the penultimate paragraph of Section 5 we include MODIS/AERONET comparisons of the retrieved Ångström coefficient over both KAUST Campus and Abu Al Bukhoosh. We do not include SEVIRI in this analysis due to the fact that the inferrence of the Ångström coefficient is not a validated product. MODIS Ångström coefficient values seem to be biased high aginst the AERONET values, consistent with what has been reported before by Levy et al. (2003, JGR). MODIS Ångström coefficient values are greater over Abu Al Bukhoosh than over KAUST, indicative of the more industrial environment of the Abu Al Bukhoosh site on an oil platform. The MODIS positive bias is not however due to its overestimate of the Ångström coefficient, since the AOD-scaling using the Ångström coefficient is negligibly different between the MODIS and AERONET values. In fact the MODIS overestimate of Ångström coefficient actually implies an underestimate of the scaled-AOD at 630 nm.

9) "It might be instructive to compare the Ångström exponent of dust over the Red Sea from east and west sources. Likewise, it may be useful to do the same thing when comparing dust over the Red Sea with that over the Persian Gulf."

In the final paragraph of Section 6 we now compare the Ångström coefficients at the basin-scale between the Red Sea and the Persian Gulf. Interestingly, while the mean Ångström coefficient over the Red Sea is 0.67, over the Persian Gulf it is distinctly higher, at 0.96. This is a signifier of the more heavily polluted urban-industrial environment of the Persian Gulf, a major centre for global oil extraction.

10) "The fonts on some of the plots are a bit small and should be increased in size."

The font sizes on several of the plots have now been increased by a couple of points.

---

## Author Response (AR2)

**Responses to reviews of 'Satellite retrievals of dust aerosol over the Red Sea and the Persian Gulf, 2005-2015'**

We thank the reviewers again for their helpful comments, and also the co-editor, Rolf Müller, for his useful comments. Below is a short response to these comments and a mark-up version of the manuscript to track the changes.

"l. 11,12: I would not state that Modis and Severi "make" retrievals. References: according to the ACP style references should be abbreviated."
Thank you for these recommendations, changes to this effect have been implemented.

"Reviewer 2. 3) Maybe a reference would be good here that discusses the advantage of using the larger spectral range of MODIS. 4) Maybe a reference would be good here that discusses the benefits of MISR's multiangle measurements."
Diner et al. (2005) has been included as a reference mentioning the benefits of MISR's multiangle viewing capabilities. An extra sentence has been included for the MODIS spectral range, citing again the Levy et al. (2003) paper to point out how MODIS uses these extra channels, as a strength of its' retrieval algorithm.

[revised manuscript text omitted]